# Circulating trophoblast cell clusters for early detection of placenta accreta spectrum disorders

Yalda Afshar[1,13], Jiantong Dong[2,3,13], Pan Zhao[4], Lei Li[2,5], Shan Wang[2,5], Ryan Y. Zhang[2], Ceng Zhang[2], Ophelia Yin[1], Christina S. Han[1,6], Brett D. Einerson[7], Tania L. Gonzalez[8], Huirong Zhang[4], Anqi Zhou[2], Zhuo Yang[2], Shih-Jie Chou[2], Na Sun[2], Ju Cheng[2], Henan Zhu[2], Jing Wang[2], Tiffany X. Zhang[2], Yi-Te Lee[2], Jasmine J. Wang[9], Pai-Chi Teng[9], Peng Yang[2], Dongping Qi[2], Meiping Zhao[3], Myung-Shin Sim[10], Ruilian Zhe[4], Jeffrey D. Goldstein[11], John Williams III[8], Xietong Wang[5], Qingying Zhang[2,12], Lawrence D. Platt[1,6], Chang Zou[4,14✉], Margareta D. Pisarska[8,14✉], Hsian-Rong Tseng[2,14✉] & Yazhen Zhu[2,14✉]

Placenta accreta spectrum (PAS) is a high-risk obstetrical condition associated with significant morbidity and mortality. Current clinical screening modalities for PAS are not always conclusive. Here, we report a nanostructure-embedded microchip that efficiently enriches both single and clustered circulating trophoblasts (cTBs) from maternal blood for detecting PAS. We discover a uniquely high prevalence of cTB-clusters in PAS and subsequently optimize the device to preserve the intactness of these clusters. Our feasibility study on the enumeration of cTBs and cTB-clusters from 168 pregnant women demonstrates excellent diagnostic performance for distinguishing PAS from non-PAS. A logistic regression model is constructed using a training cohort and then cross-validated and tested using an independent cohort. The combined cTB assay achieves an Area Under ROC Curve of 0.942 (throughout gestation) and 0.924 (early gestation) for distinguishing PAS from non-PAS. Our assay holds the potential to improve current diagnostic modalities for the early detection of PAS.

[1] Division of Maternal Fetal Medicine, Department of Obstetrics and Gynecology, David Geffen School of Medicine, University of California, Los Angeles, Los Angeles, CA, USA. [2] California NanoSystems Institute, Crump Institute for Molecular Imaging, Department of Molecular and Medical Pharmacology, University of California, Los Angeles, Los Angeles, CA, USA. [3] Beijing National Laboratory for Molecular Sciences, MOE Key Laboratory of Bioorganic Chemistry and Molecular Engineering, College of Chemistry and Molecular Engineering, Peking University, Beijing, China. [4] Clinical Medical Research Center, The First Affiliated Hospital of Southern University of Science and Technology, The Second Clinical Medical College of Jinan University, Shenzhen People's Hospital, Shenzhen, Guangdong, China. [5] Department of Obstetrics and Gynecology, Shandong Provincial Hospital Affiliated to Shandong University, Jinan, Shandong, China. [6] Center for Fetal Medicine and Women's Ultrasound, Los Angeles, CA, USA. [7] Division of Maternal-Fetal Medicine, Department of Obstetrics & Gynecology, University of Utah Health, Salt Lake City, UT, USA. [8] Department of Obstetrics and Gynecology, Cedars-Sinai Medical Center, Los Angeles, CA, USA. [9] Samuel Oschin Comprehensive Cancer Institute, Cedars-Sinai Medical Center, Los Angeles, CA, USA. [10] Departments of Computational Medicine & Medicine, David Geffen School of Medicine, University of California, Los Angeles, Los Angeles, CA, USA. [11] Department of Pathology and Laboratory Medicine, Ronald Reagan Medical Center, David Geffen School of Medicine, University of California, Los Angeles, Los Angeles, CA, USA. [12] Department of Obstetrics, Obstetrics and Gynecology Hospital, Fudan University, Shanghai, China. [13] These authors contributed equally: Yalda Afshar, Jiantong Dong. [14] These authors jointly supervised this work: Hsian-Rong Tseng, Yazhen Zhu, Margareta D. Pisarska, Chang Zou. ✉email: zou.chang@szhospital.com; margareta.pisarska@cshs.org; hrtseng@mednet.ucla.edu; yazhenzhu@mednet.ucla.edu

Placenta accreta spectrum (PAS) disorders, including placenta accreta, placenta increta, and placenta percreta, are the consequences of abnormal implantation[1], or aberrant invasion and adherence of placental trophoblasts into the uterine myometrium[2]. PAS is associated with significant maternal morbidity because the post-delivery placenta of the fetus does not spontaneously separate and can lead to severe hemorrhage, often leading to an emergency hysterectomy, blood transfusion, and intensive care unit admission[3,4]. Safe and optimal care of pregnant women with PAS depends on antenatal diagnosis[3,5]. Current diagnostic modalities for PAS, including serum analytes, ultrasonography, and magnetic resonance imaging (MRI), are effective but not always conclusive, and some options are not readily available in low resource settings[6–8]. Even in specialized diagnostic units in the United States, around one-third[9] to half[10] of PAS cases remain undiagnosed during pregnancy. Thus, there is a crucial need to develop novel technologies to improve the antenatal diagnosis of PAS throughout gestation and several blood-based biomarkers—such as plasma protein signatures[11], cell-free fetal DNA and cell-free placental mRNA[12]—have been explored for this purpose. Timely detection and diagnosis of PAS provide opportunities to improve prenatal care and minimize maternal and neonatal morbidity by planning delivery in a tertiary care center with a coordinated team. This has implications both from the individual's risk-stratification and from a broader public health perspective[13–16]. The most important risk factors for PAS are placenta previa (when the placenta implants low and overlays the cervix) and prior cesarean deliveries[17]. Given the rising rate of cesarean deliveries, there has been a concomitant 100-fold increase in the incidences of PAS disorders since the 1950s, with a current prevalence of 1 in 500 pregnancies[18]. A noninvasive approach for the early detection of PAS is valuable to inform providers and women of their high-risk pregnancy in all health systems, especially in low resource and rural settings without sub-specialists trained in ultrasound[15,16,19].

Circulating trophoblasts (cTBs) are placenta-derived trophoblast cells, predominantly of the extra-villous trophoblast (EVT) type, which shed into the maternal circulation during placental implantation and development[20,21]. Even though rare in numbers, cTBs can be enriched from the maternal circulation. We and others have demonstrated these cells can be used for genetic testing and potentially used as an alternative for noninvasive prenatal testing (NIPT)[20,22–25]. EVT's function is to migrate and invade at the maternal-fetal interface for normal implantation and placentation; however, when dysfunctional, abnormal invasion occurs, it can lead to PAS[17,26]. This leads to potential increased cTBs present in the maternal circulation and a means to detect abnormal placental invasion noninvasively. Exploring the utility of cTB enumeration as noninvasive biomarkers for the assessment of excessive EVT invasion may be a promising diagnostic solution to detect PAS throughout gestation.

Our group pioneered the concept of "NanoVelcro" Chips[27,28], in which immunoaffinity agent-coated nanostructured substrates confer improved "stickiness" to the devices, allowing for selective capture of cTBs from pregnant women[25], as well as other types of rare cells, e.g., circulating tumor cells (CTCs) from cancer patients[29–31].

In this work, our initial goal is to explore the use of NanoVelcro Chips to detect increased cTBs in maternal circulation as a result of abnormal migration and invasion leading to PAS compared to normal placentation. However, during a pilot study, we discovered the presence of aggregates of cTBs in clusters, now known as "clustered cTBs" in PAS. To best preserve the intrinsic properties (i.e., morphology and size distribution) of cTB-clusters while retaining the capture performance of the device, we carry out comprehensive optimization to enable simultaneous detection of both single and clustered cTBs. Using the optimized NanoVelcro Chips, we enumerate single and clustered cTBs, as well as cTB-clusters in a cohort of 168 pregnant women with clinically confirmed PAS, placenta previa and normal placentation. Control studies are performed in 15 healthy non-pregnant female donors. We are able to demonstrate that the counts of single and clustered cTBs, as well as cTB-clusters in PAS are significantly higher than those in non-PAS groups, and the combination of single and clustered cTBs, as well as cTB-clusters can be used to distinguish PAS from normal placentation and/or placenta previa with excellent diagnostic performance in training, validation, and test cohorts throughout gestation, particularly early in gestation. In addition, the EVT origin of cTBs captured on NanoVelcro Chips is verified by well-validated immunocytochemistry (ICC) markers and further confirmed by the detection of trophoblast-specific genes, including those of EVT origin, using reverse transcription Droplet Digital polymerase chain reaction (RT-ddPCR). Enumeration of single and clustered cTBs, as well as cTB-clusters can be used for the noninvasive early detection of PAS, holding great promise to improve current diagnostic modalities for PAS detection.

## Results

**Discovery of clustered cTBs in PAS using NanoVelcro Chips.** To test our hypothesis that cTB counts in maternal circulation are elevated in PAS compared to normal placentation, we conducted a pilot study to capture cTBs using the NanoVelcro Chip (Fig. 1a and Supplementary Fig. 1), which is composed of two functional components, i.e., an anti-EpCAM-grafted silicon nanowire substrate (SiNWS) and an overlaid PDMS chaotic mixer. Conceptually, NanoVelcro Assays work analogously like Velcro™: the target cell surfaces covered with nanoscale cell-surface components (a.k.a., microvilli) and the substrate embedded with nanostructures can be regarded as the upper and lower strips of Velcro fastener, respectively. When a cTB contacts the substrate, the microvilli on cTBs' surfaces entangle with the nanostructures on the NanoVelcro Chips, introducing increased surface contact areas to facilitate immunoaffinity-mediated cTB capture. Following the previously published procedure for preparation of SiNWS[27], Ag nanoparticle-templated wet etching was employed to introduce densely packed silicon nanowires with a high aspect ratio (diameters = 100–200 nm, lengths = 5–10 μm) onto lithographically patterned silicon wafers. N-hydroxysuccinimide/maleimide chemistry was then adopted to covalently conjugate streptavidin onto the surfaces of SiNWS. Before cTB-capture studies, biotinylated anti-EpCAM were grafted onto SiNWS to confer the specificity to recognize and enrich single and clustered cTBs in blood samples. Here, NanoVelcro Chips pre-coated with anti-EpCAM were utilized to isolate and enumerate cTBs in pregnant women with normal placentation ($n = 2$), placenta previa ($n = 3$), and PAS ($n = 5$). A 4-color ICC protocol[25] was developed for the immunofluorescent staining of the captured cTBs. In addition to a conventional cTB marker, i.e., CK7, human leukocyte antigen (HLA)-G—a major histocompatibility tissue-specific antigen that is normally expressed in EVTs[32,33]—was used to verify the identity of the cTBs and enhance the specificity of this assay. Fluorescence microscopy imaging was adopted to distinguish cTBs (DAPI+/CK7+/HLA-G+/CD45–, Fig. 1b) from background white blood cells (WBCs) (DAPI+/CK7–/HLA-G–/CD45+) immobilized on SiNWS. In pregnant women with PAS, in addition to single cTBs, we observed a phenomenon of clustered cTBs (cTBs present in the cTB-clusters, which were defined as an aggregation of two or more cTBs). The clustered cTBs shared similar cytomorphology and similar immuno-phenotype with the single cTBs. In this pilot study, no clustered cTBs were

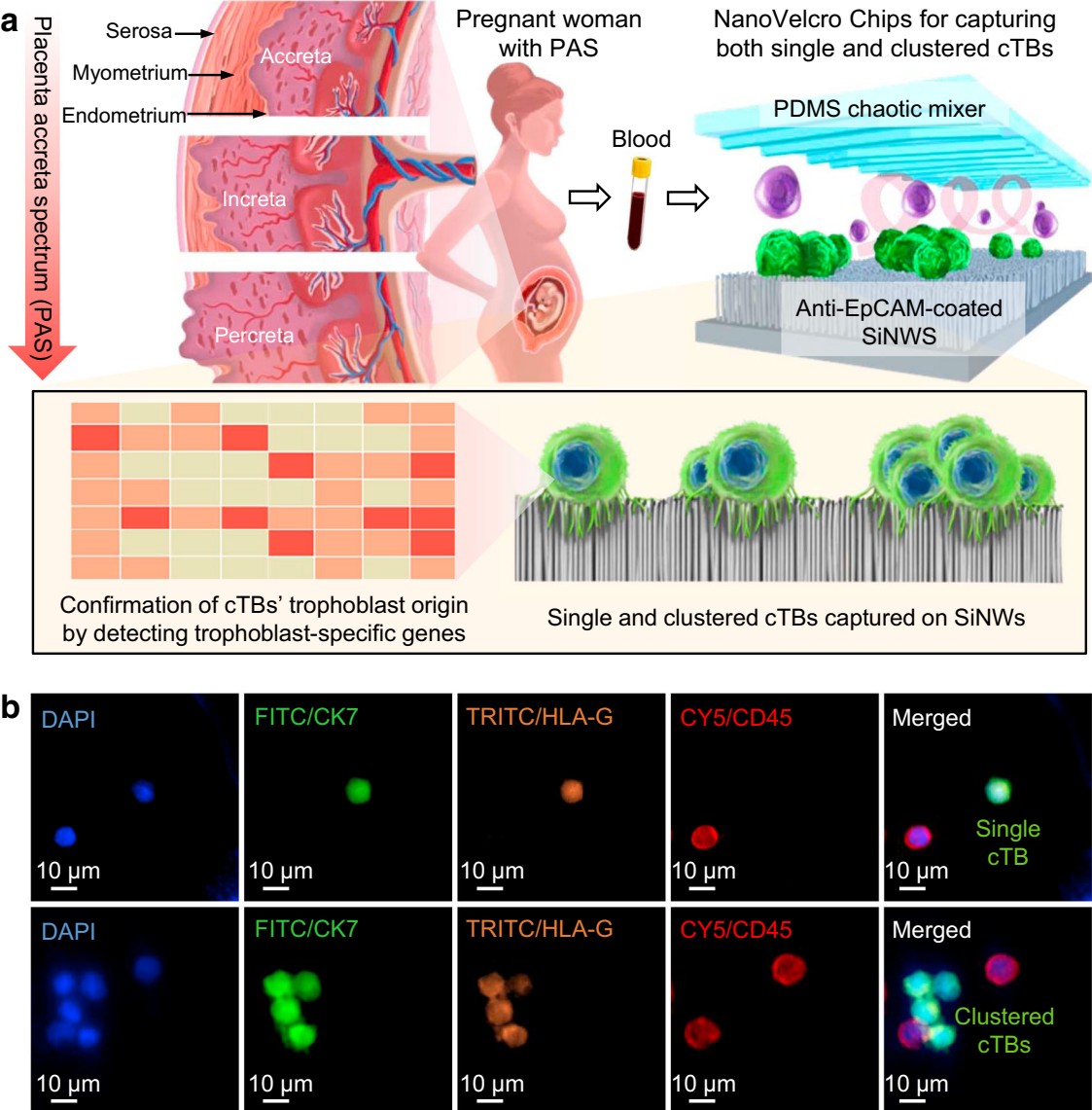

**Fig. 1 NanoVelcro Chips for detecting single and clustered circulating trophoblasts (cTBs) in placenta accreta spectrum (PAS) disorder. a** Abnormal invasion and adherence of placental trophoblasts into the uterine myometrium, classified into placenta accreta, increta, and percreta based on the severity of the disorder. During implantation and placentation, a small number of cTBs sheds from the placenta into the maternal circulation. NanoVelcro Chip, composed of an overlaid polydimethylsiloxane (PDMS) chaotic mixer and an anti-EpCAM-coated silicon nanowire substrates (SiNWS), was adopted to capture both single and clustered cTBs (colored in green) in maternal blood, allowing for noninvasive detection of PAS disorder. The trophoblast origin of the cTBs was confirmed by detecting trophoblast-specific genes and immunocytochemistry (ICC) staining on the captured cTBs. **b** Representative micrographs of ICC staining on a single cTB and clustered cTBs (DAPI+/CK7+/HLA-G+/CD45–) captured by NanoVelcro Chips. Blue: DAPI stained nuclei; green: FITC stained CK7; orange: TRITC stained HLA-G; red: CY5 stained CD45. Scale bar, 10 μm. Data are representatives of five independent assays.

detected in pregnant women with normal placentation or placenta previa.

**Optimization for isolating both single and clustered cTBs.** To best preserve the intrinsic properties (including intactness and size distribution) of cTB-clusters while capturing both single and clustered cTBs, we optimized NanoVelcro Chips according to the general workflow depicted in Fig. 2a. Initially, we obtained single and clustered cTBs by culturing JEG-3 cells (a trophoblast cell line) under sphere-forming conditions[34]. In a typical sphere-forming JEG-3 cell sample, the majority (67.7%) of cells are present in the form of multidirectional clusters. For the convenience of cell counting by fluorescence microscopy, single and clustered JEG-3 cells in the mixture were labeled with a

3,3'-dioctadecyloxacarbocyanine perchlorate (DiO) green fluorescence dye. We prepared clustered cTB blood sample models by spiking a healthy non-pregnant female donor's peripheral blood mononuclear cells (PBMCs) (isolated from 2-mL blood) with single and clustered JEG-3 cells. The clustered cTB blood sample models were run through NanoVelcro Chips, followed by the nuclear staining with 4′,6-diamidino-2-phenylindole (DAPI), then microscopy imaging and enumerating. We first examined how the flow rates (0.2, 0.5, 1.0, and 2.0 mL h$^{-1}$) affected the efficiencies of capturing single JEG-3 cells and JEG-3 clusters of varying cell numbers. The data summarized in Fig. 2b suggests that 0.5 mL h$^{-1}$ is the optimal flow rate with an average capture efficiency of 94% for single JEG-3 and 93–100% for clustered JEG-3 (of cell numbers ranging between 2 and >20). Overall, we found that (i) NanoVelcro Chips exhibited better capture

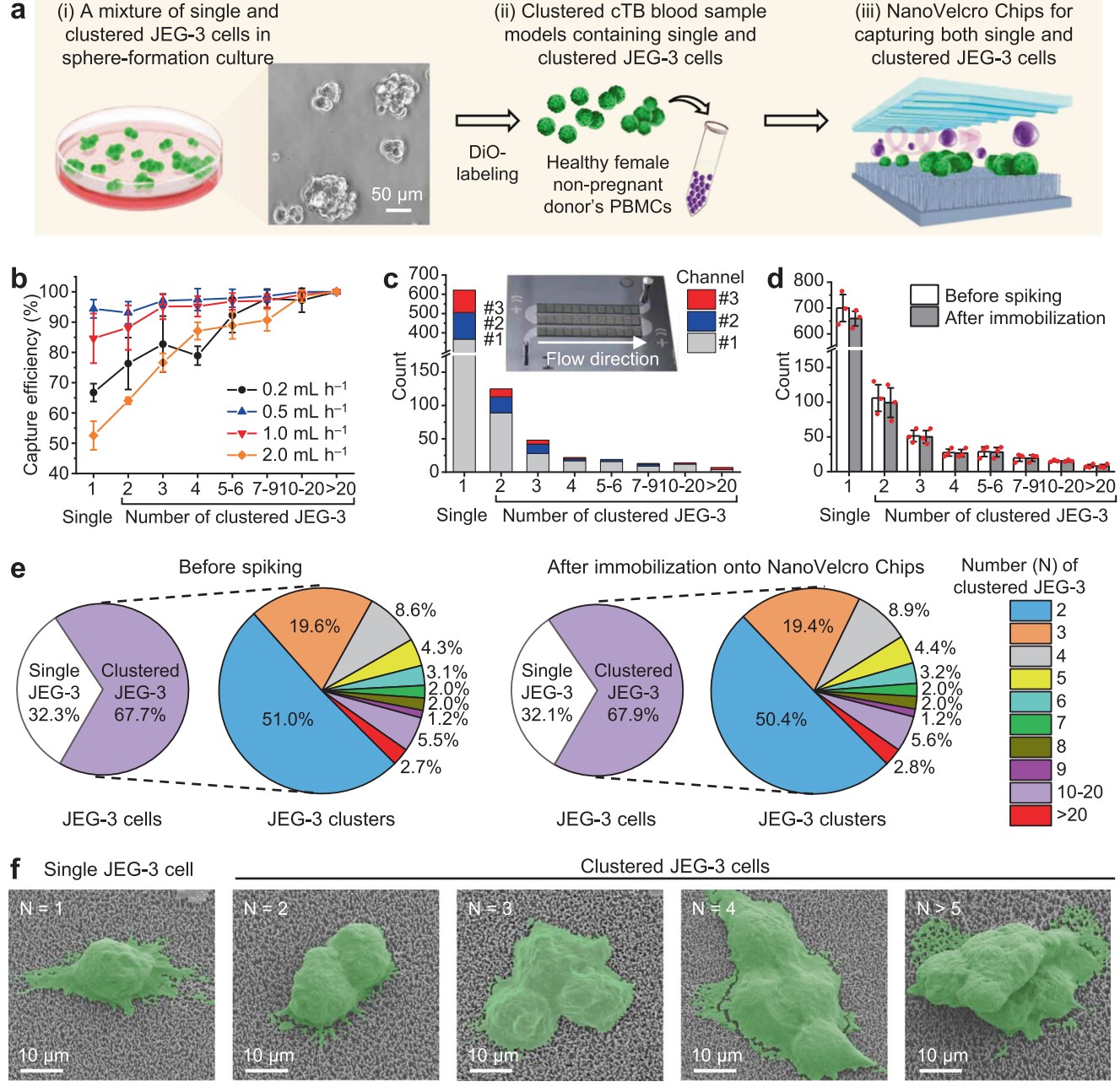

**Fig. 2 Optimization and characterization of NanoVelcro Chips for capturing single and clustered cTBs in clustered cTB blood sample models. a** A general workflow developed for optimization of NanoVelcro Chips for capturing both single and clustered JEG-3 cells. (i) A mixture of single and clustered JEG-3 cells was prepared by culturing under a sphere-formation condition. (ii) Both single and clustered JEG-3 cells were labeled with 3,3'-dioctadecyloxacarbocyanine perchlorate (DiO) and spiked into healthy non-pregnant female donor's peripheral blood mononuclear cells (PBMCs) to prepare clustered cTB blood sample models. (iii) These samples were used for the optimization of NanoVelcro Chips to capture both single and clustered JEG-3 cells. Scale bar, 50 μm. **b** The performance of capturing single JEG-3 cells and JEG-3 clusters of varying cell numbers were studied at flow rates of 0.2, 0.5, 1.0, and 2.0 mL h⁻¹. Data are presented as means ± SD of three independent assays. **c** Distribution of single JEG-3 cells and JEG-3 clusters along the three microchannels of NanoVelcro Chips were studied at the optimal flow rate of 0.5 mL h⁻¹. Data are from one independent experiment. Inset: a photograph of NanoVelcro Chip showing the connected three channels. **d** Side-by-side comparison of the distributions of single JEG-3 cells and JEG-3 clusters of varying cell numbers characterized before spiking and after immobilization onto NanoVelcro Chips. Data are presented as means ± SD of three independent assays. Error bars represent SD. **e** Representative pie charts showing the proportions of single and clustered JEG-3 cells, as well as proportions of JEG-3 clusters of varying cell numbers, before spiking (left) and after (right) immobilization. Data are from one representative independent experiment. **f** SEM images of the single JEG-3 cell (N = 1) and clustered JEG-3 cells (N = 2, 3, 4, and >5) captured on the SiNWS of NanoVelcro Chips. Cells are colored in green. Scale bar, 10 μm. Data are representatives of three independent assays. Source data are provided in the Source data file.

performance for larger clusters than that observed for single cells and smaller clusters, and (ii) higher flow rates negatively impact the overall capture performance.

**NanoVelcro Chip introducing negligible perturbation to cTBs.** At the optimal flow rate of $0.5 \, \text{mL h}^{-1}$, we evaluated the spatial distribution of single and clustered JEG-3 cells along the three channels on each NanoVelcro Chip. As shown in Fig. 2c, 59% (367/622), 22% (139/622), and 19% (116/622) of single JEG-3 cells were captured in the first, second, and third channels, respectively. 71% (175/248), 19% (47/248), and 10% (26/248) of clustered JEG-3 were captured in the first, second, and third channels, respectively. The predominant distribution in the first two channels (81–90%) suggests that NanoVelcro Chips have sufficient channel length to capture both single and clustered JEG-3 cells. Finally, we studied how the capture process in NanoVelcro Chips could perturb the intrinsic properties (i.e., intactness and size distribution) of JEG-3 clusters. Figure 2d shows a side-by-side comparison of the distribution of single JEG-3 cells and JEG-3 clusters of varying cell numbers before spiking and after immobilization onto NanoVelcro Chips. This is also depicted in the representative pie charts in Fig. 2e. Therefore, the distribution observed after immobilization is similar to that before spiking, suggesting that NanoVelcro Chips introduces negligible perturbations to the intrinsic properties of cTB-clusters. To better elucidate how the Velcro-like operating mechanism[27,30] facilitates immunoaffinity-mediated capture of single and clustered JEG-3 cells onto SiNWS, scanning electron microscopy (SEM) imaging was employed to characterize the interfaces between single and clustered JEG-3 cells and SiNWS after capture. Fig. 2f shows representative SEM images of a single JEG-3 cell ($N = 1$) and clustered JEG-3 cells ($N = 2, 3, 4,$ and $>5$) captured on SiNWS of NanoVelcro Chips. We observed the entangled interactions between long microvilli of JEG-3 cells and densely packed silicon nanowires on SiNWS, i.e., the characteristic "Velcro-like" interactions at the interfaces between single and clustered JEG-3 cells and SiNWS. In conclusion, NanoVelcro Chips introduced negligible perturbations to the morphology of single and clustered JEG-3 cells, suggesting that the Velcro-like operating mechanism, originally developed to facilitate immunoaffinity-mediated capture of single cTBs, can also be effectively adopted for immunoaffinity-mediated capture of clustered cTBs.

**Single and clustered cTBs isolated from maternal blood.** Using the above optimized experimental conditions, NanoVelcro Chips were employed to detect and enumerate single and clustered cTBs, as well as cTB-clusters in clinical samples following the streamlined workflow (see "Methods"). Fig. 3 depicts our clinical study design. Among the 171 eligible pregnant women recruited in this study, three subjects were excluded due to fetal genetic/congenital anomalies or technical failure. We collected blood samples from 168 pregnant women and 15 non-pregnant women in four cohorts, (i) PAS cohort: prenatally suspected and subsequently pathologically confirmed PAS patients ($n = 65$, mean age = 36 years old (yo)); (ii) placenta previa cohort: clinically diagnosed placenta previa patients ($n = 59$, mean age = 35 yo); (iii) normal placentation cohort: pregnant women with clinically confirmed normal placentation ($n = 44$, mean age = 37 yo); and (iv) healthy non-pregnant female donors ($n = 15$, mean age = 29 yo). The demographic information of these cohorts is provided in Table 1 and Supplementary Table 1. Characterization and enumeration were performed by an investigator blinded to all clinical information. For each blood sample, PBMCs were obtained from 2 mL of whole blood (via gradient centrifugal

depletion of red blood cells) and then processed through Nano-Velcro Chips. After performing the 4-color ICC[25] to stain DAPI, CK7, HLA-G, and CD45, single and clustered cTBs (DAPI+/CK7+/HLA-G+/CD45−) were identified from background WBCs (DAPI +/CK7−/HLA-G−/CD45+) under a fluorescence microscope. Representative micrographs of a single cTB, as well as clustered cTBs in different sizes of cTB-clusters comprising of 2, 8, and 15 cells isolated from blood samples of pregnant women with PAS, are shown in Fig. 4a, b. Additional images of single cTBs, as well as clustered cTBs in different sizes of cTB-clusters, are provided in Supplementary Fig. 2. The morphology of single cTBs is usually round and smooth with diffuse nuclear DAPI staining and uniform cytoplasmic CK7 staining. cTB-clusters are characterized as an aggregation of two or more cTBs, and the clustered cTBs are the cTBs present in a cTB-cluster. Both the single and clustered cTBs exhibit cytoplasmic and membrane HLA-G staining. The overall size of cTB-clusters ($n = 279$) ranges from 7 μm to 210 μm (Supplementary Fig. 3a, b) depending on the configuration and numbers of cTBs in each cluster. The distribution and proportion of cTB-clusters of varying numbers of cells are summarized in Supplementary Fig. 3c, d. Despite their size differences, cTB-clusters are composed of small round cTBs with a high level of homogeneity.

**Single and clustered cTBs as a putative biomarker for detecting PAS.** Blood samples from a final cohort of 168 pregnant women were subjected to NanoVelcro Chip assay and analyzed according to the combination of single and clustered cTBs, as well as cTB-clusters. The final cohort included 65 PAS, 59 placenta previa, and 44 normal placentation. The enumeration results (cTB count per 2 mL of blood) of single cTBs (blue bar) and clustered cTBs (orange bar) from each blood sample, are summarized in Fig. 5a. Overall, single cTBs are detected in the majority of pregnant women, with a detection rate of 98, 85, and 86% in the groups of PAS, placenta previa, and normal placentation, respectively (Fig. 5a inset). Notably, the detection rates of clustered cTBs (i.e., cTBs present in cTB-clusters) in the PAS group (86%) were statistically significantly higher (Chi-square test, $p < 0.0001$) than the placenta previa group (22%) and the normal placentation group (14%). Of the 6 PAS samples without clustered cTBs, 3 samples were from women with a focal accreta confirmed intraoperatively, a less severe phenotype of PAS. This result raises the question of the correlation between cTB enumeration and severity of the disease. Therefore, we first compared the counts of single and clustered cTBs, as well as cTB-clusters among the subtypes of PAS (i.e., accreta versus increta and percreta). Results summarized in Supplementary Fig. 4. indicated that despite a trend showing increased numbers of cTBs or cTB-clusters with more severe disease, the comparison is not statistically significant ($p > 0.05$) in the current study. cTBs were not detected in healthy non-pregnant female donors. We then compared the counts of single and clustered cTBs, as well as cTB-clusters among the three study groups (i.e., PAS, Previa, and Normal) in Fig. 5b–d. Significantly higher counts of single and clustered cTBs, as well as cTB-clusters were observed in the PAS group compared to those found in the Previa and Normal groups, suggesting the potential role of cTBs and cTB-clusters in distinguishing PAS from placenta previa, and normal placentation.

To integrate the enumeration results of single and clustered cTBs, as well as cTB-clusters into a statistically robust prediction model, we first screened each variable to determine if it served as a statistically significant univariate predictor of PAS status. Receiver operating characteristic (ROC) analysis was conducted to assess the diagnostic performance of the single and clustered cTBs, as well as cTB-clusters to distinguish pregnant women with PAS from those with placenta previa and normal placentation. ROC curves summarized in Supplementary Fig. 5. demonstrated

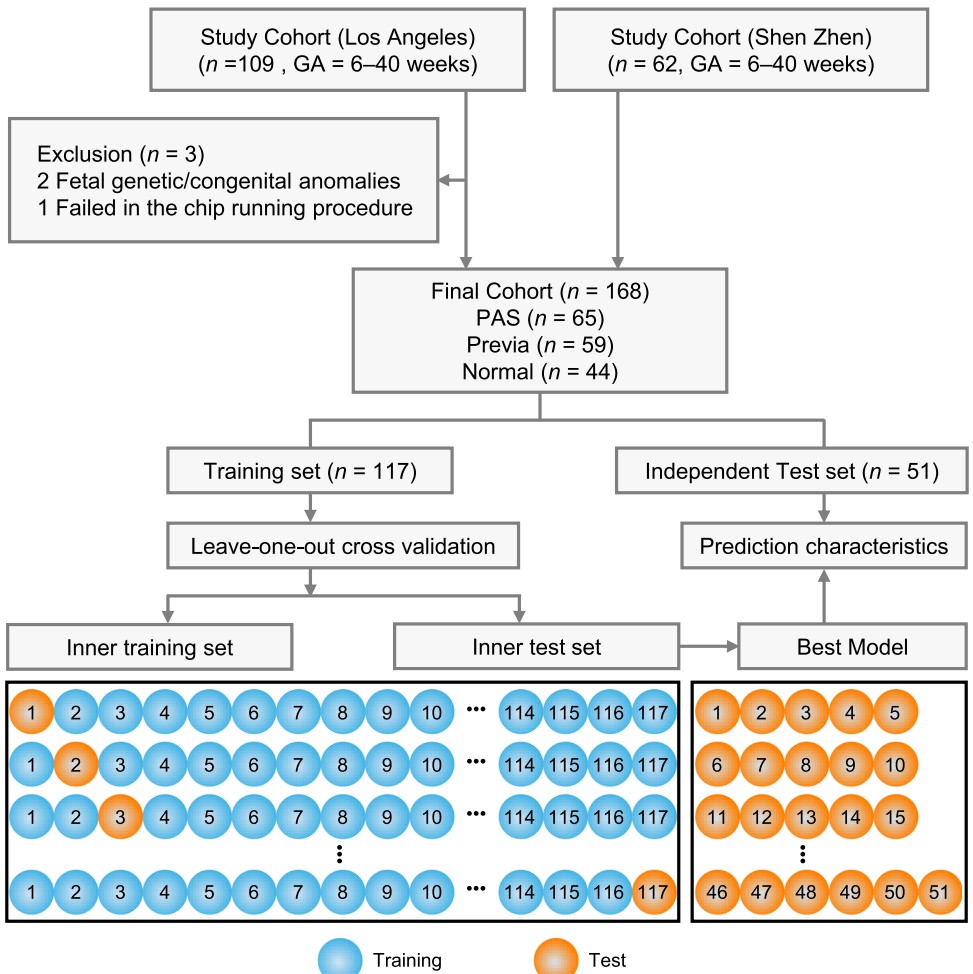

**Fig. 3 Clinical study design flowchart depicting the recruitment and exclusions from the study cohort.** Blood samples from 109 and 62 eligible study subjects gestational age (GA, 6–40 weeks) were collected and processed in Los Angeles, USA, and Shenzhen, China, respectively. After 3 samples were excluded due to fetal genetic/congenital anomalies or technical failure, blood samples from a final cohort of 168 pregnant women were subjected to NanoVelcro Chip assay and analyzed according to the combination of single and clustered cTBs, as well as cTB-clusters.

that all of single and clustered cTBs, as well as cTB-clusters met this selection criterion. We then conducted a stepwise multivariate logistic regression model (Supplementary Table 2) to combine the single and clustered cTBs, as well as cTB clusters for differentiating PAS from non-PAS (placenta previa plus normal placentation) followed by the leave-one-out cross validation, and applied this logistic regression model to an independent test set (Fig. 3). Fig. 6a–c summarizes the diagnostic performance of the combined cTB assay for distinguishing PAS from non-PAS in the training set, after leave-one-out cross validation, and in the independent test set with areas under the curve (AUC) of 0.947 (sensitivity = 88.9%, specificity = 87.5%), 0.946 (sensitivity = 88.9%, specificity = 87.5%), and 0.926 (sensitivity = 90.0%, specificity = 90.3%), respectively. We observed that the PAS prevalence is higher in Shenzhen cohort (47.5%) than the USA cohort (33.6%) in this study. We next calculated the positive predictive value (PPV) and negative predictive value (NPV) for the cTB assay in the subpopulations of USA cohort (PPV = 85.3%, NPV = 90.4%) and Shenzhen cohort (PPV = 92.9%, NPV = 90.9%) as well as all cohorts (PPV = 83.8%, NPV = 92.0%). These data are summarized in Supplementary Fig. 6.

To explore whether the cTB assay improves the prediction of PAS if combined with ultrasound and other clinical risk factors listed in Table 1, such as previous cesarean delivery (CD),

maternal age, maternal body mass index (BMI), in vitro fertilization (IVF), gravidity and parity, multivariate logistic regression analysis was used to evaluate and select significant clinical predictors. The results showed that in addition to the cTB assay, ultrasound is the most statistically significant contributor associated with PAS ($p < 0.0001$, Supplementary Table 2). We then constructed a stepwise multivariate logistic regression model to see if the combination of the cTB assay with ultrasound improves prediction of PAS. As the comparison of ROC curves showed in Fig. 6d, the combination of cTB assay along with ultrasound achieved an AUC of 0.978, which outperformed the cTB assay alone (with AUC of 0.978 versus 0.942, $p = 0.0523$, not significant), or ultrasound alone (with AUC of 0.978 versus 0.866, $p < 0.0001$).

Both single and clustered cTBs can be detected throughout gestation. The counts of single and clustered cTBs, as well as cTB-clusters based on GA for each group are displayed in Supplementary Fig. 7. There was no statistically significant difference between earlier GA and late GA in the counts of single and clustered cTBs, as well as cTB-clusters in PAS and normal placentation. The statistically significant decrease was only observed for single cTBs in placenta previa. To explore whether the cTB assay performs differently in varying GA, we then conducted logistic regression models (Supplementary Table 2)

**Table 1 Clinical information for pregnant women ($n = 168$) enrolled in the study.**

| Characteristics | PAS | Previa | Normal | p value[a] |
|---|---|---|---|---|
| Total number (n) | 65 | 59 | 44 | |
| Median maternal age (range)-yo | 36 (23–44) | 35 (19–54) | 37 (22–45) | 0.8772 |
| Pre-pregnancy BMI[b] (range) | 24 (16–49) | 23 (17–32) | 24 (19–44) | 0.4858 |
| IVF[c]-n (%) | | | | 0.5830 |
| Yes | 14 (21.5) | 11 (18.6) | 15 (34.1) | |
| No | 51 (78.5) | 48 (81.4) | 29 (65.9) | |
| Gravidity-n (%) | | | | 0.0180 |
| 1 | 8 (12.3) | 7 (11.9) | 15 (34.1) | |
| 2 | 11 (16.9) | 16 (27.1) | 15 (34.1) | |
| ≥3 | 46 (70.8) | 36 (61.0) | 14 (31.8) | |
| Parity-n (%) | | | | 0.0294 |
| 0 | 10 (15.4) | 13 (22.0) | 20 (45.5) | |
| 1 | 26 (40.0) | 19 (32.2) | 21 (47.7) | |
| ≥2 | 29 (44.6) | 27 (45.8) | 3 (6.8) | |
| Previous CD[d]-n (%) | | | | 0.0013 |
| 0 | 15 (23.1) | 21 (35.6) | 32 (72.7) | |
| 1 | 30 (46.1) | 20 (33.9) | 11 (25.0) | |
| ≥2 | 20 (30.8) | 18 (30.5) | 1 (2.3) | |

[a]PAS versus non-PAS.
[b]BMI: body mass index (kg m⁻²).
[c]IVF: In vitro fertilization.
[d]CD: cesarean delivery. Comparisons are evaluated using Mann–Whitney test or the Chi-square test. All tests are two-sided without adjustments.

transcriptome datasets through the Human Protein Atlas, (https://www.proteinatlas.org/humanproteome/tissue/placenta)[35–37]. As summarized in Supplementary Fig. 8, we initially identified 91 placenta enriched genes that had the highest gene expression compared to other tissues (≥4-fold higher mRNA expression). Subsequently, to increase specificity, we selected genes that were detected either only in placenta (5 genes) or in less than one-third of other tissue types (46 genes). We then used a high tissue specificity (TS) score (TS > 28), which is determined by fold-change between the expression in placenta to tissue with the second-highest expression level, to further increase placental specificity of our placenta specific genes. Since brain and blood cells have unique datasets in the Human Protein Atlas (The Brain Atlas and The Blood Atlas, respectively), only genes that had low/absent expression in brain or WBCs (NX ≤ 1) were selected. These selection steps resulted in a panel of 12 genes that are highly specific for the placenta (Supplementary Fig. 8). To identify genes specific to the trophoblast population of the placenta, which is the population in the maternal circulation[20,21], we utilized publicly available single-cell RNA sequencing data of the placenta (i.e., GSE89497[38]). Of the identified 12 placenta specific genes, we selected highly expressed genes in trophoblasts and excluded those also highly expressed in villous stromal cells. Trophoblast candidate genes that were highly expressed throughout gestation were selected to broaden utilization. To minimize differences due to fetal sex, trophoblast candidate genes were not sexually dimorphic[39]. The resulting 7 trophoblast-specific genes were further confirmed in another publicly available single-cell RNA sequencing dataset of trophoblasts, GSE9773[40] (Supplementary Fig. 8). The final 7 trophoblast-specific genes include *chorionic somatomammotropin hormone (CSH)1, CSH2, pappalysin (PAPPA)2, pregnancy-specific beta-1-glycoprotein (PSG)1, PSG2, PSG3, PSG11* (Supplementary Table 4).

We validated the trophoblast specificity of our 7 genes using trophoblast cells from the placental tissue of PAS compared to WBCs from healthy non-pregnant female donors (Fig. 7a). Hematoxylin and Eosin (H&E) staining and HLA-G immunohistochemistry staining of placenta tissue ensured that trophoblast cells are identified and dissected for RT-ddPCR (Supplementary Fig. 9). Heat maps of placenta-derived gene signatures obtained from 4 placenta samples and 4 non-pregnant female donor WBC specimens (Fig. 7a, heat maps) demonstrates that all 7 trophoblast-specific genes are highly expressed in the trophoblast within the placenta from PAS patients and absent in WBCs. Trophoblast-specific gene expression was studied in cTBs captured on NanoVelcro Chips from 21 pregnant women including 11 with PAS and 10 with normal placentation. PBMCs were obtained from 2 mL of whole blood and processed through the NanoVelcro Chips. RNA was extracted from single and clustered cTBs captured on the NanoVelcro Chips, followed by RT-ddPCR for the trophoblast-specific gene detection (Fig. 7a). As depicted in the heat maps (Fig. 7b), signals of the 7-validated trophoblast-specific genes were identified in both pregnant women with PAS and normal placentation. The primary copy numbers were log10-transformed for each gene. Of the 7 trophoblast-specific genes, all had significantly higher expression in PAS than those in normal placentation, except *PSG2* (Fig. 7c). These results further confirmed the trophoblast origin of the single and clustered cTBs enriched by NanoVelcro Chips.

and comparison of ROC analysis on the subpopulations of pregnant women at earlier GA (<24 weeks) and late GA (≥ 24 weeks) to distinguish PAS from non-PAS, respectively. The cTB assay in the subpopulations can differentiate PAS from non-PAS regardless of GA. Again, the cTB assay can improve prediction of PAS if combined with ultrasound in both earlier GA and late GA. In the earlier GA population (Fig. 6e), the combination of cTB assay along with ultrasound achieved an AUC of 0.976, which outperformed the cTB assay alone (with AUC of 0.976 versus 0.924, $p = 0.1713$, not significant), or ultrasound alone (with AUC of 0.976 versus 0.826, $p = 0.0015$). In the late GA population (Fig. 6f), the combination of cTB assay along with ultrasound achieved an AUC of 0.979, which outperformed the cTB assay alone (with AUC of 0.979 versus 0.961, $p = 0.2548$, not significant), or ultrasound alone (with AUC of 0.979 versus 0.884, $p = 0.0014$). These results show that the potential of cTB assay for the detection of PAS can be used throughout gestation, and the cTB assay is also reproducible in tight GA windows of both earlier GA and late GA.

In this study, the numbers of single ($p = 0.0103$, $r_s = 0.197$) and clustered cTBs ($p = 0.0151$, $r_s = 0.187$), as well as cTB-clusters ($p = 0.0151$, $r_s = 0.187$) are correlated with a previous CD. However, they are not correlated with other known risk factors for PAS including maternal age, maternal BMI, IVF, or gravidity and parity. Among all 63 cases of PAS, there are 43 (66%) patients who have both a previous CD and a previa and 5 (8%) patients have neither a previous CD nor a previa (Supplementary Table 3), which are the two most important risk factors for PAS.

**Confirming the trophoblast origin of the cTBs.** We further confirmed the trophoblast origin of the cTBs by detecting trophoblast-specific genes. To identify trophoblast-specific genes throughout gestation, we used publicly available human placenta

## Discussion

In this study, we engineered and optimized nanostructure-embedded microchips (i.e., NanoVelcro Chips) to fulfill an unmet clinical need for the early detection of PAS. The capacity of Nano-Velcro Chips to enrich and detect both single and clustered cTBs

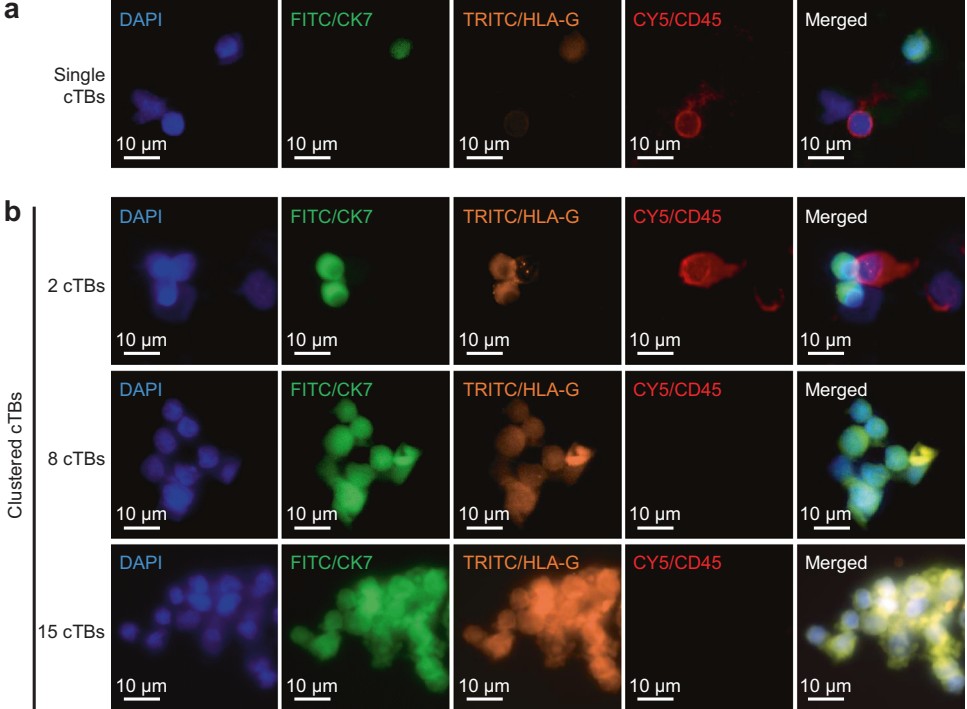

**Fig. 4 Characterization and enumeration of single and clustered cTBs isolated from blood samples collected from pregnant women.** Representative immunofluorescent images of **a** single and **b** clustered cTBs in different sizes of cTB-clusters shown at ×40 magnification. Blue: DAPI stained nuclei; green: FITC stained CK7; orange: TRITC stained HLA-G; red: CY5 stained CD45. Scale bar, 10 μm. Data are representatives of four independent assays.

simultaneously enables accurate enumeration of single and clustered cTBs, as well as cTB-clusters. The counts of single and clustered cTBs, as well as cTB-clusters were significantly higher in women with PAS than placenta previa and normal placentation throughout gestation, and also in the subpopulations of both earlier GA and late GA. The enumeration of cTBs and cTB-clusters holds great promise for the early detection of PAS, and augments ultrasound screening, with implication to low resource settings. We discovered and characterized cTB-clusters, a striking characteristic that has never been described for PAS. Although cTBs have been detected in maternal blood using multiple technologies[20,22–25], and clustered cTBs were observed when detecting single cTBs using immuno-magnetic cell sorting methods[24,41], the majority of current cTB detecting technologies were focused on NIPT. To our knowledge, none of them have been developed to enrich single and clustered cTBs simultaneously for detecting PAS.

Current screening paradigms for PAS include clinical history risk stratification combined with 2D ultrasonography[42], with MRI as an adjunct to ultrasound in cases of more severe placental invasion or cases with posterior placentation[4]. However, the imaging-based antenatal diagnosis of PAS disorders remains subjective, with accuracy dependent on the operator, similar to limitations seen for fetal anomaly screening[7]. Moreover, it was reported that MRI is often misleading when used as an adjunct to ultrasound in the management of PAS[6], and MRI is expensive and requires expertise that is rarely available in lower resource settings. Recent population studies have shown that half to two-thirds of cases of PAS disorders remain undiagnosed before delivery[43,44], highlighting the crucial need to develop new technologies for prenatal detection. Our study demonstrates a promising noninvasive technology for detecting PAS that does not rely on imaging instruments or expertise by taking advantage of the in vitro diagnostic value of NanoVelcro Chips, capable of accurately enumerating single and clustered cTBs, as well as cTB-clusters.

The combination of single and clustered cTBs, as well as cTB-clusters can differentiate PAS from placenta previa and normal placentation during pregnancy with excellent diagnostic performance throughout gestation, but most importantly, the cTB assay can also be applied early in gestation. cTBs have been identified to decline with increasing GA in normal placentation[24,41]. In this study, no significant difference was observed between the earlier and late GA in counts of single, and clustered cTBs, as well as cTB-clusters in PAS and non-PAS groups except the single cTBs in the placenta previa group, which may be explained by the high rate of resolution of previas later in gestation. The cTB assay to distinguish PAS from non-PAS performed well throughout gestation and it also performed reproducibly in both earlier and late GA. The cTB assay provides the opportunity for the non-invasive detection of PAS earlier in gestation, with significant potential to expedite early intervention, including referral of these pregnancies to tertiary care centers and Centers of Excellence for PAS, which provides the opportunity to improve clinical outcomes[45].

In this study, we pioneered the use of single and clustered cTBs, as well as cTB-clusters enriched by NanoVelcro Chips for detecting PAS. Significantly increased numbers of single and clustered cTBs, as well as cTB-clusters, were observed in PAS in contrast to non-PAS. The uniqueness of our NanoVelcro Chips stems from the use of nanostructured substrates, which allow for enhanced local topographic interactions between the nanos-tructured substrates and nanoscale cellular surface components (e.g., microvilli), resulting in vastly improved cell-capture affinities[30,31]. This unique mechanism of NanoVelcro Chips makes them well-suited for capturing single cTBs and even more so for capturing cTB-clusters. In addition, our NanoVelcro Chips introduce negligible perturbations to the cTB-clusters during the capture process. The prevalence of cTB-clusters with more than two cTBs attached to each other makes them more visible and less biased for enumeration. Given the streamlined workflow

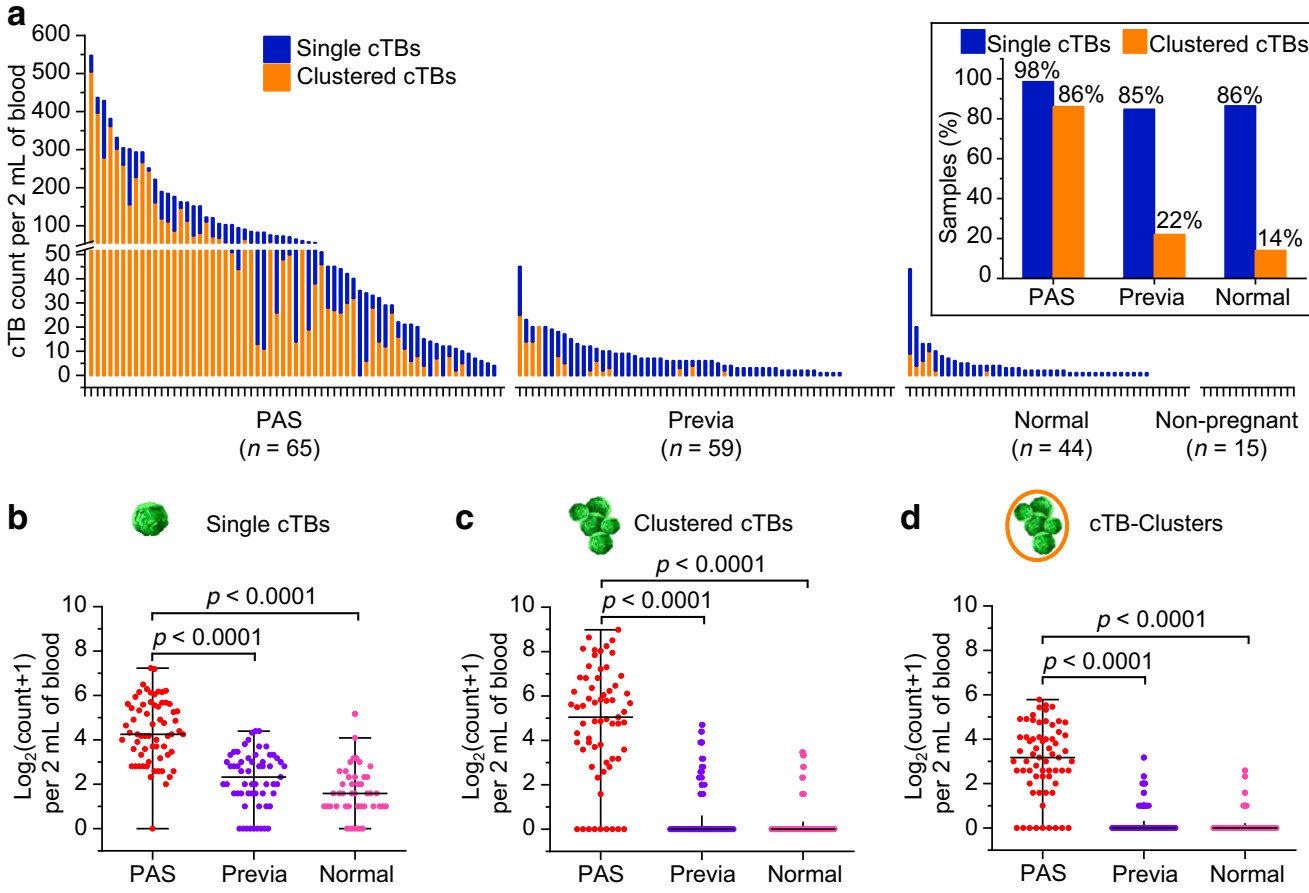

**Fig. 5 Comparison of single and clustered cTBs, as well as cTB-clusters in PAS, placenta previa, and normal placentation. a** Counts of single and clustered cTBs per 2 mL of blood for all participants enrolled in the study ($n = 183$). Pregnant women ($n = 168$) are divided into three groups (i.e., PAS, placenta previa, and normal placentation) based on clinical diagnosis and sorted within groups based on total cTB count (single cTB count plus clustered cTB count). Non-pregnant indicates the control cohort of healthy non-pregnant female donors ($n = 15$). The inset shows the percentages of pregnant women with detectable single and clustered cTBs in PAS, placenta previa, and normal placentation. Counts of: **b** single cTBs, **c** clustered cTBs, and **d** cTB-clusters per 2 mL of blood for pregnant women with PAS ($n = 65$), placenta previa ($n = 59$), and normal placentation ($n = 44$) are shown. Counts are log2-transformed. The two shorter horizontal lines denote the 25–75% interquartile ranges (IQR) and the longer horizontal lines in between denote the median. Data are expressed as mean ± SE for **b** single cTBs: PAS ($4.4 \pm 0.2$), placenta previa ($2.2 \pm 0.2$), normal placentation ($1.6 \pm 0.2$); **c** clustered cTBs: PAS ($4.7 \pm 0.3$), placenta previa ($0.6 \pm 0.2$), normal placentation ($0.3 \pm 0.1$); and **d** cTB-clusters: PAS ($3.0 \pm 0.2$), placenta previa ($0.3 \pm 0.1$), normal placentation ($0.2 \pm 0.1$). Significant differences between different groups are evaluated using one-way ANOVA with Bonferroni's multiple comparisons test (two-sided with adjustment). The $p$ values for **b** single cTBs, **c** clustered cTBs, and **d** cTB-clusters (PAS versus previa; PAS versus normal)) are all <0.0001. Source data are provided in the Source data file.

developed for capturing cTBs and cTB-clusters in NanoVelcro Chips, we envision scaling up this cost-effective assay in a large-scale multicenter validation study.

cTBs have been identified in maternal circulation as far back as 1893 when multinucleate fetal trophoblasts were reported at postmortem examinations of lungs from women dying as a result of eclampsia[46]. Unlike the cTBs found in the pulmonary vasculature postmortem, our cTB-clusters are in the venous system, which requires passage through capillaries. Thus it remains to be determined if this clustering is unique to the type of trophoblast that has extra invasive properties leading to PAS and travels as cTB-clusters[24], similar to tumor cells that are more aggressive and have the capability to enter the venous system as large CTC-clusters[47]. In previous studies[25,48], cTBs have been demonstrated to be of fetal origin as they carry both paternal and maternal alleles. In this study, we further confirmed they are of trophoblast origin, similar to recent studies that identified cTBs to be EVTs of placental origin[20,21]. Moreover, HLA-G expression[49] in-situ on both single and clustered cTBs supports they are all of the same immune-phenotype and derived from an EVT cell type.

We note that this study has some limitations. Due to the relatively low incidence of PAS, our internal and external validation study was conducted in a relatively small cohort though patients were from different hospitals. Interestingly, of the 6 PAS samples without clustered cTBs and with the lowest counts of single cTBs (Fig. 5a, bottom 6 in PAS group), 3 samples were from pregnant women with small areas of focal accreta, which is a less severe form of placenta accreta. This result raises the possibility of the correlation between cTB enumeration and severity of the disease. However, our results of the stratification of PAS showed no significant difference between the accreta group versus increta/percreta group in the counts of single and clustered cTBs, as well as cTB-clusters. More evidence from large-scale validation studies are required to address this question. We lack longitudinal follow-up to be able to interpret the implications of dynamic changes of single and clustered cTBs across gestation in pregnant women who are at higher risk of developing PAS. Future clinical validation studies will require larger cohorts.

In summary, this is the first description of the prevalence of cTB-clusters in pregnant women with PAS. To enable accurate

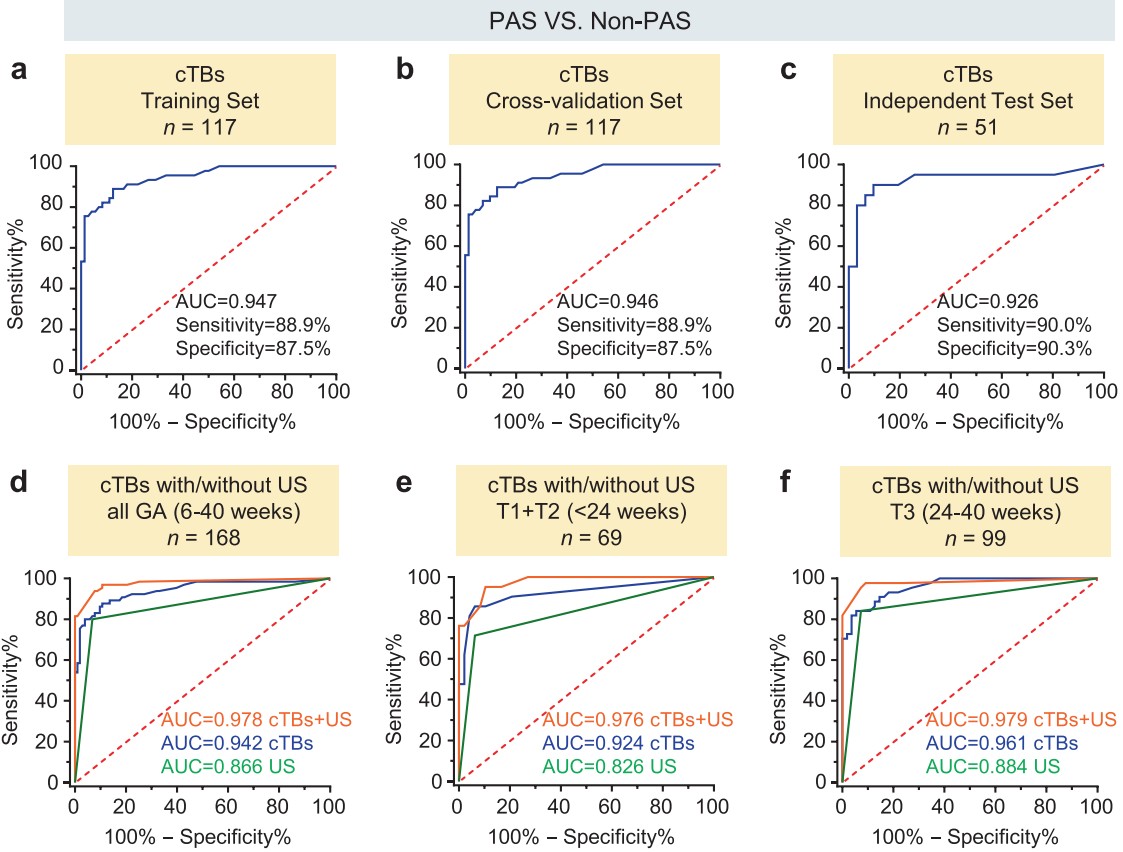

**Fig. 6 Receiver operating characteristic (ROC) curves of cTB assay with/without ultrasound.** ROC curves of cTB assay analyzed in **a** the training set, **b** after leave-one-out cross validation, and **c** independent test set for distinguishing PAS from non-PAS during all gestational age (GA, 6–40 weeks). Area under the curves (AUC) with the sensitivity and specificity of the assays at the optimal cutoffs are listed for each graph. Comparison of ROC curves of cTB assay with or without ultrasound (US) for distinguishing PAS from non-PAS in **d** all GA (6–40 weeks), **e** earlier GA (The first and second trimester, T1 + T2 < 24 weeks), and **f** late GA (T3 ≥ 24 weeks). AUC for each diagnostic model is listed for each ROC curve. Source data are provided in the Source data file.

cTB and cTB-cluster enumeration for detecting PAS, we engineered and optimized a nanostructure-embedded microchip that can isolate both single and clustered cTBs efficiently. We demonstrate that enumeration of single and clustered cTBs, as well as cTB-clusters isolated by the NanoVelcro Chips can be used for noninvasive detection of PAS throughout gestation, but most importantly can be used earlier in gestation, which holds great promise to augment current diagnostic paradigms for detecting PAS. We envision that the combination of cTBs and cTB-clusters captured on the NanoVelcro Chips for detecting PAS early in gestation will enable a promising quantitative assay to serve as a noninvasive test and also as a complement to ultrasonography to improve diagnostic accuracy for PAS early in gestation.

## Methods

**Study design**. This observational cohort study protocol was approved by the Institutional Review Boards (IRB) of University of California, Los Angeles (UCLA) (UCLA IRB#13-001264), University of Utah Health (MTA-2018-1091), Cedars-Sinai Medical Center (CSMC) (CSMC IRB #Pro00006806 and Pro00008600) and Shenzhen People's Hospital (LL-KY-2019608). All samples from pregnant women and healthy non-pregnant female donors were obtained according to protocols approved by the IRB. Written informed consent was obtained from all the participating subjects. Pregnant women aged from 18 to 45 years old with singleton intrauterine pregnancies, and GA between 6 and 40 weeks were eligible for inclusion. Women were excluded if they are diagnosed with known or suspected aneuploidy, fetal genetic/congenital anomalies or blood draw was not possible. Samples were collected between December 2017 and January 2021 during prenatal care visits. Pregnant women were classified as normal placentation, placenta previa (without placenta accreta), and PAS. PAS or placenta previa was defined and

diagnosed according to current American College of Obstetricians and Gynecologists (ACOG) and Society for Maternal-Fetal Medicine (SMFM) guidelines[4] as well as FIGO consensus guidelines[7]. The primary diagnostic modality for antenatal diagnosis was a maternal-fetal medicine obstetrics ultrasound. PAS was confirmed by histopathological diagnosis after delivery after reviewed by pathologists with expertise in gynecological and perinatal pathology as part of routine clinical care for those patients who underwent a hysterectomy. The patients who did not have a hysterectomy were confirmed PAS during the cesarean delivery[7]. An intraoperative or clinical diagnosis of PAS is made in accordance with the International FIGO classification of PAS based on general classification and grading[3] (Supplementary Table 3). Normal placentation had no clinical evidence of pregnancy complications or any fetal abnormalities. The pilot study was conducted at UCLA. NanoVelcro Chips were optimized using clustered cTB blood sample models and then used to isolate and detect cTBs and cTB-clusters from clinical samples for detection of PAS at both UCLA (samples collected in USA) and Shenzhen (samples collected in China). All blood samples collected at CSMC or University of Utah Health were sent to UCLA for NanoVelcro Chip assay on the day of blood draw. The sample size was calculated according to the AUC comparison between our assay and the clinical ultrasound using the paired DeLong's test. A sample size of 128 (51 cases of PAS and 77 cases of control, the ratio of sample sizes in negative/positive groups is 3/2) is expected to have 80% power to detect the difference between the AUCs for cTB assay versus ultrasound, assuming AUC = 0.920 for our cTB assay, AUC = 0.800 for ultrasound, when a correlation between the assays of 0.5 was assumed. The power was obtained for a two-sided test at 0.05 significance level. All laboratory samples were assayed by investigators blinded to the clinical status of the subjects.

**Preparation of single and clustered JEG-3 cells**. The human choriocarcinoma cell line JEG-3 was acquired from the American Type Culture Collection (ATCC, USA) and cultured in DMEM supplemented with 10% fetal calf serum in a humidified atmosphere of 5% $CO_2$ at 37 °C. The cell line was tested and found negative for Mycoplasma contamination. A floating mixture of single and clustered JEG-3 cells were generated under a sphere-forming condition[34] and used to test the

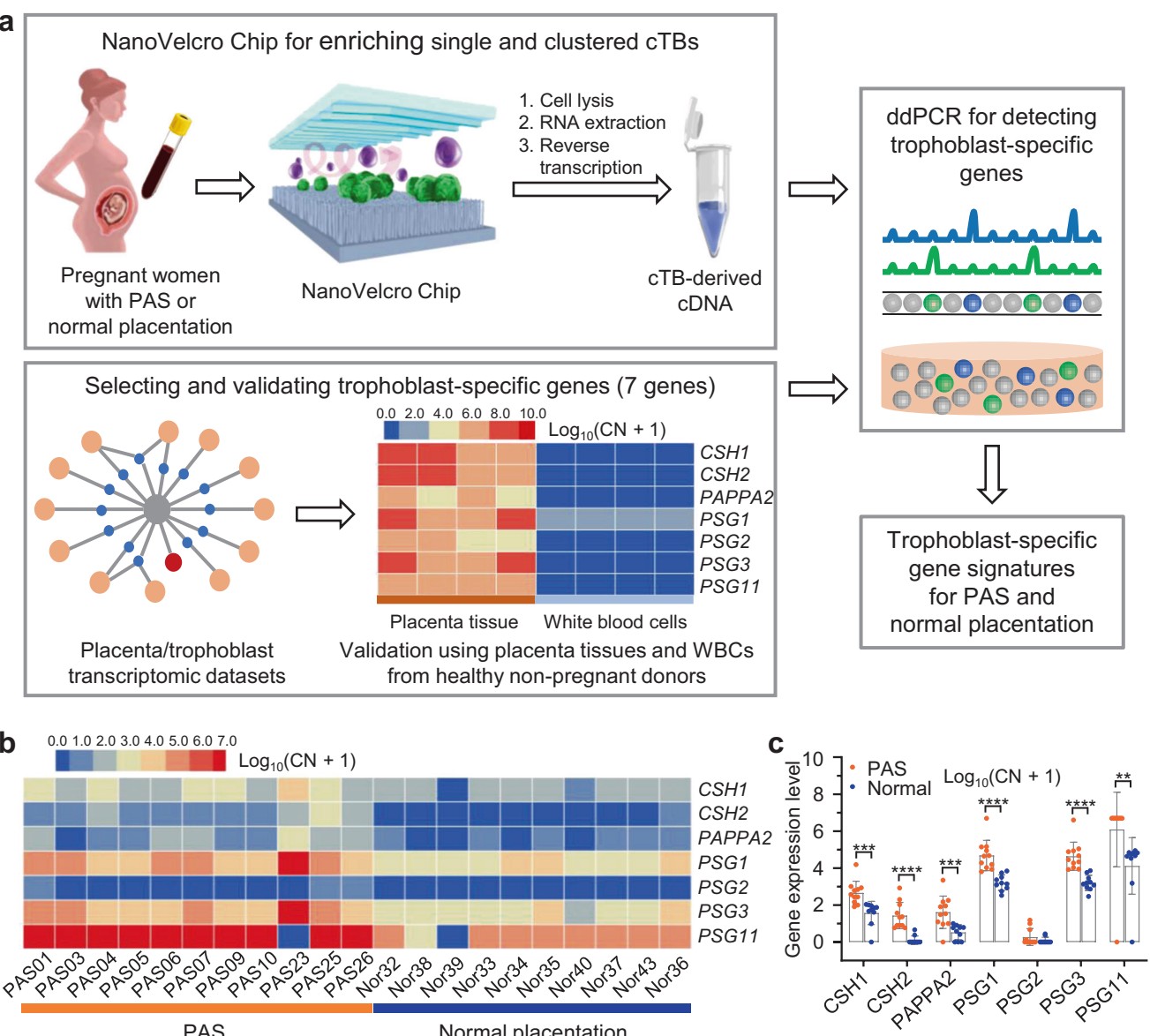

**Fig. 7 RT-ddPCR assay for detection of trophoblast-specific genes in the cTBs captured by NanoVelcro Chips confirming trophoblast cells of placental origin. a** Schematic illustrating the general workflow for detecting trophoblast-specific genes in cTBs captured by NanoVelcro Chips. The trophoblast-specific genes were selected from placenta transcriptome databases, validated using trophoblast cells from the placental tissue of PAS ($n = 4$ biologically independent samples) compared to white blood cells (WBCs) from healthy non-pregnant female donors ($n = 4$ independent experiments). **b** Heat maps depicting relative signal intensities for gene expression of 7 trophoblast-specific genes in the cTBs enriched from pregnant women with PAS ($n = 11$) and normal placentation ($n = 10$). **c** Differences in gene expression for 7 trophoblast-specific genes in the cTBs enriched from pregnant women with PAS ($n = 11$) and normal placentation ($n = 10$). Primary copy numbers (CN) are log10-transformed for each gene and False Discovery Rate (FDR) is controlled for multiple comparisons. Data are presented as means ± SD. Multiple T-tests are used to compare the differences of trophoblast-gene expression in different groups. All tests are two-sided with adjustments. The adjusted $p$ value ($q$ value) for each gene (PAS versus normal) is as follows: 0.0004 (CSH1), <0.0001 (CSH2), 0.0008 (PAPPA2), <0.0001 (PSG1), 0.0671 (PSG2), <0.0001 (PSG3), 0.0074 (PSG11). ****$q$ < 0.0001, ***$q$ < 0.001, **$q$ < 0.01, *$q$ < 0.05. Clinical data for patients are listed in Table 1 and Supplementary Table 1. Source data are provided in the Source data file.

performance of NanoVelcro Chips for capturing both single and clustered JEG-3 cells. Briefly, logarithmic phase JEG-3 cells were digested in 0.25% trypsin, and once cells detached, digestion was terminated with serum-free culture media (SFM) composed of DMEM/F12 (1:1) basal medium (Hyclone, USA) supplemented with 20 ng mL$^{-1}$ human epidermal growth factor (EGF) (PeproTech, USA), 20 ng mL$^{-1}$ basic fibroblast growth factor (bFGF) (PeproTech, USA), 0.4% bovine serum albumin (BSA) (Sigma, USA), 4 µg mL$^{-1}$ insulin, 1:50 B27 supplement (Gibco/Invitrogen, Australia) and 100 U mL$^{-1}$ penicillin. A single-cell suspension of 5000–20000 JEG-3 cells was seeded in each well of a low adhesion 6 well plate. The medium was replaced every other day. Cell sphere-formation was observed by inverted microscopy; when cell spheres expanded 50-fold, the supernatant was collected, centrifuged at $300 \times g$ for 5 min, and the cell spheres were washed with

phosphate buffer solution (PBS) twice. The live single and clustered JEG-3 cells were characterized by acquiring an optical microscope image before dye labeling. Then, the mixture of single and clustered JEG-3 cells was pre-stained with Vybrant™ DiO green fluorescent dye, and the PBMCs isolated from healthy non-pregnant female donor's whole blood were pre-stained with Vybrant™ DiD red fluorescent dye (Invitrogen, USA). The cell pre-staining process was performed in a serum-free culture medium at 37 °C for 1 h for cell-labeling before the spiking study. Excess cell-labeling dye was removed by centrifuging the labeled suspension at $300 \times g$ for 5 min and washed with PBS twice. After dye labeling, the cell mixture was resuspended with PBS. A 2.5-µL mixture of single and clustered JEG-3 cells was deposited on an ultralow-attachment culture dish[50]. The population of single and clustered JEG-3 cells was characterized by acquiring fluorescence microscope

images before spiking. After that, the mixture of single and clustered JEG-3 cells was re-pipetted and spiked into PBMCs to prepare a clustered cTB blood sample model for the immobilization onto a NanoVelcro Chip. The culture dish was reimaged to account for the mixture of single and clustered JEG-3 cells that remained attached to the surface. By post-processing these microscope images, single JEG-3 cells and clustered JEG-3 cells within each cluster that were ultimately spiked into female donor's whole blood could be accurately enumerated.

**NanoVelcro Chip optimization**. The clustered cTB blood sample models containing both single and clustered JEG-3 cells were incubated with biotin-conjugated anti-EpCAM for 30 min at room temperature, and excess antibody was removed by PBS washing and centrifuging the suspension at $300 \times g$ for 5 min. The samples were resuspended in PBS (200 μL) and injected to the NanoVelcro Chips at flow rates of 0.2, 0.5, 1.0, and 2.0 mL h$^{-1}$, respectively. The single and clustered JEG-3 cells captured in NanoVelcro Chips were fixed with 4% formaldehyde (PFA, Electron Microscopy Sciences) in PBS (200 μL). After nuclear staining with DAPI, all the stained chips were scanned and imaged under a fluorescence microscope (Nikon 90i). The single and clustered JEG-3 cells captured on the chips were then counted.

**SEM sample preparation and imaging**. A mixture of single and clustered JEG-3 cells was captured onto the NanoVelcro Chips. After separating the PDMS tops from the chips, the single and clustered JEG-3 cells were first fixed in 4% PFA for 1 h. After fixation, cells were dehydrated in 50, 70, 80, 95, and 100% ethanol (200 proof) solutions successively for 15 min each. The samples were dried overnight and then sputter-coated with gold at room temperature. Prepared samples were imaged using a Zeiss Supra 40VP SEM at an accelerating voltage of 10 keV.

**Clinical sample collection and blood processing**. Blood samples were collected in the Department of Obstetrics and Gynecology, David Geffen School of Medicine, UCLA; Los Angeles and the Center for Fetal Medicine and Women's Ultrasound; Los Angeles; Department of Obstetrics and Gynecology, Cedars-Sinai Medical Center, Los Angeles; Department of Obstetrics & Gynecology, University of Utah Health, Salt Lake City, UT, USA; Department of Obstetrics and Gynecology, Shenzhen People's Hospital, China. Clinical information was abstracted from all subject charts. Peripheral blood samples were collected in 10 mL acid citrate dextrose (ACD) vacutainer tubes and were processed on the same day. The PBMCs were isolated in 15 min by using SepMate™ PBMC Isolation Tubes (STEMCELL Technologies, USA) and aliquoted following our previously published protocol[51]. PBMCs from 2 mL of whole blood were run through the NanoVelcro Chips using the optimized protocol. The streamlined workflow is composed of the following steps: (i) capturing cTB cells/clusters onto NanoVelcro Chips (30 min), (ii) overnight standard immunostaining, and (iii) imaging/counting by a fluorescent microscope (30 min).

**Immunostaining and identification of cTBs**. The samples were run through NanoVelcro Chips under optimized conditions. After 4% PFA fixation, the immobilized cells were subjected to 4-color ICC staining with DAPI (#422801, BioLegend, USA), anti-cytokeratin 7 antibody (Rabbit polyclonal IgG) (#ab53123, abcam, USA), anti-HLA-G antibody (4H84) (#ab52455, abcam, USA), and anti-CD45 monoclonal antibody (YAML501.4) (#MA5-17687, Thermo Fisher Scientific, USA) for identification of cTBs (DAPI+/CK7+/HLA-G+/CD45−) and cTB-clusters. Immobilized cells were imaged using the Nikon Ni fluorescence microscope with NIS-Element imaging software (Nikon Eclipse Ti2). An automatic scan was carried out under ×20 magnification with DAPI, FITC, TRITC, and Cy5 channels corresponding to nuclear, CK7, HLA-G, and CD45 staining, respectively. Micrographic features of candidate cTBs were reviewed to ensure consistency with epithelial as opposed to the hematologic origin. When analyzing the multi-channel ICC micrographs, WBCs were defined as round/ovoid cells (DAPI+/CK7−/HLA-G−/CD45+); and cTBs were defined as round/ovoid cells (DAPI+/CK7+/HLA-G+/CD45−).

**Pathological examination of placenta tissues**. Pathological examination including Hematoxylin and Eosin (H&E) and immunohistochemistry (IHC) staining of the placenta tissues obtained from PAS patients was performed independently by pathologists with expertise in gynecological and perinatal pathology at UCLA. All of the placenta tissues were fixed in 10% neutral formalin for 24–48 h and embedded in paraffin according to the standard operating procedure (SOP) for tissue in the pathology department at UCLA. Serial 4 μm-thick tissue sections from formalin-fixed paraffin-embedded (FFPE) blocks were cut and mounted on poly-L-lysine coated glass slides. Standard IHC staining on 4 μm-thick tissue sections was performed on Ventana Benchmark ULTRA Slide Stainer according to a protocol optimized for the HLA-G antibody. The IHC analysis for HLA-G was performed to differentiate between villous trophoblasts (cytotrophoblasts and syncytiotrophoblasts) and EVTs. Positive HLA-G staining confirmed the identity of EVTs.

**Trophoblast-specific gene validation on placenta tissues**. Trophoblast-specific gene expression in placenta tissue was performed to validate the selected

trophoblast-specific gene panel. The unstained FFPE placenta tissue slides were deparaffinized and macrodissected to enrich the trophoblasts in the placenta tissues. The target trophoblastic areas in each slide were identified and marked under microscopy in the corresponding H&E and IHC stained slides from the same FFPE block by pathologists. Total RNA was extracted from the enriched placenta tissues using Qiagen (Dusseldorf, Germany) RNeasy FFPE kit. Then the complementary DNA (cDNA) was synthesized using a Thermo Scientific Maxima H Minus Reverse Transcriptase Kit according to the manufacturer's protocols. cDNA was then tested for trophoblast-specific gene transcripts using duplex ddPCR in each tube with two fluorescence filters (i.e., FAM and VIC). Predesigned Taqman assays (Thermo Fisher Scientific) containing primers and probes for each gene (Supplementary Table 5) were used in the ddPCR following the manufacturer's protocols. ddPCR experiments were performed on a QX200 system (Bio-Rad Laboratories, Inc.) according to the manufacturer's protocols. Data were analyzed using the QuantaSoft™ software to quantify the corresponding copy numbers of gene transcripts detected in each assay.

**Trophoblast-specific gene detection on the isolated cTBs**. NanoVelcro Chips were used for isolating the single and clustered cTBs from pregnant women. To extract RNA from the single and clustered cTBs captured on NanoVelcro Chips, we performed on-chip lysis of cells by introducing 600 μL of TRIzol solution (Zymo Research, USA) and 600 μL of anhydrous ethanol (Sigma-Aldrich) sequentially through the NanoVelcro Chips. The effluent solution was collected in a 2.0 mL RNase-free Eppendorf tube at the same time. Then, RNA was purified using a Direct-zol™RNA MicroPrep Kit (Zymo Research). cDNA was synthesized using a Thermo Scientific Maxima H Minus Reverse Transcriptase Kit according to the manufacturer's instructions. cDNA was then tested for the trophoblast-specific gene using duplex ddPCR in each tube with two fluorescence filters (i.e., FAM and VIC). RT-ddPCR experiments were performed using the same protocol as that used for the placenta tissues.

**Statistical analysis**. Comparisons of cTB enumeration data among different groups were evaluated using one-way ANOVA after log2-transformation. Comparisons of the enumeration data between earlier GA and late GA were evaluated using Mann–Whitney test. Chi-square test was used for comparison of the detection rates of samples for cTBs and cTB-clusters in each group. Comparisons in Table 1 were evaluated using Mann–Whitney test or the Chi-square test. The Spearman's rank-order correlation was used to analyze the correlation of the cTB enumeration and demographic information. Multiple T-tests were used to compare the differences of trophoblast-gene expression in different groups. Benjamini, Krieger, and Yekutieli FDR were used for multiple testing correction across the set of genes with the maximum desired FDR of 1%. Logistic regression model, comparison of ROC curves, and single ROC curves were conducted using MedCalc 19.0.4 software. The optimal cutpoints were calculated to maximize sensitivity and specificity. Leave-one-out cross validation for the logistic regression model was conducted in R studio (version 1.4.1103-4). All the other statistical tests in this study were performed using the GraphPad Prism 9.0 software (https://www.graphpad.com/). All tests are two-sided and p-value or q-value <0.05 is considered significant.

**Reporting summary**. Further information on research design is available in the Nature Research Reporting Summary linked to this article.

## Data availability
Raw data of Figs. 2b–e, 5, 6, 7, Table 1, Supplementary Figs. 3b–d, 4, 5, 6, 7, and Supplementary Table 2, 3 are provided as a Source data file. All the other data that support the findings of this study are available within the article and its Supplementary information files. Additional data are available from the corresponding authors upon reasonable request. Source data are provided with this paper.

## Code availability
The code for leave-one-out cross validation for the logistic regression model used in this paper is provided in Supplementary Note 1.

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

## Acknowledgements

This work was supported by National Institutes of Health (U01EB026421, R01CA246304, R01CA218356, R01CA253651, R21CA235340, R21CA240887, R01CA255727, and R01HD091773) as well as the Iris Cantor-UCLA Women's Health Center Executive Advisory Board and NCATS UCLA CTSI Grant Number UL1TR000124. We thank Professor Jeffrey Gornbein from UCLA for his suggestions on statistical analysis.

## Author contributions

Y.A., J.D., M.D.P., Y.Z., and H.-R.T. designed and performed most of the research and data analysis. Y.A., C. Zou, R.Z., M.D.P., L.L., and S.W. performed the collection of blood samples and clinical information from pregnant women with assistance from C.S.H., O.Y., L.D.P., B.D.E., and X.W. J.D. and Y.Z. made schematic diagrams and figures with assistance from T.X.Z. J.D. performed device modification of NanoVelcro Chips, cell capture studies, SEM characterization, fluorescence imaging, and cell enumeration with assistance from D.Q., A.Z., Z.Y., P.Y., and N.S. J.D., R.Y.Z., P.Z., and C. Zhang performed the isolation of cTBs from blood samples using NanoVelcro Chips. Y.Z. performed the identification and enumeration of single and clustered cTBs isolated from blood samples. R.Y.Z. performed RT-ddPCR experiments with assistance from J.D. and J.W. T.L.G. performed gene selection. J.D.G. and Y.Z. reviewed the pathological tissue slides and marked the trophoblastic areas for microdissection. S.J.C. performed sphere-cell culture. J.C., H. Zhang, H. Zhu, T.X.Z., Y.-T.L., J.J.W., P.-C.T., R.Z., M.Z., J.W.III, and Q.Z. provided input, assistance, and advice on the project. Y.Z. and H.-R.T. oversaw project

execution. M.-S.S. contributed to the statistical analysis. Y.A., J.D., Y.Z., and H.-R.T. wrote the manuscript with input from O.Y. and M.D.P.

## Competing interests

Following the management plan provide by UCLA Conflict of Interest Review Committee, Dr. Hsian-Rong Tseng would like to disclose that (i) he has a financial interest in CytoLumina Technologies Corp. and Pulsar Therapeutics Corp., and (ii) the UC Regents have licensed intellectual properties invented by Dr. Tseng to CytoLumina and Pulsar. The remaining authors declare no competing interests.
