## [Peer Review File · Nature Communications]

REVIEWER COMMENTS

Reviewer #1 (Remarks to the Author):

The authors demonstrate that the counts of single, clustered and total cTBs in PAS are significantly higher than those in non-PAS groups, and the counts of cTBs can be used to distinguish PAS from normal placentation and/or placenta previa. The paper is generally clear and well prepared, but the authors should address the following comments to further clarify their work:

1) Although the EVT origin of cTBs captured on NanoVelcro Chips was verified by well-validated immunocytochemistry (ICC) markers and confirmed by the detection of trophoblast-specific genes by RT-ddPCR, but they didn't confirm the origin of CTBs from fetuses. This point should be discussed.

2) The clone number of antibodies including anti-HLA-G, anti-CK7 and the companies from which they were purchased should be added.

3) In the methods section, it was mentioned, 'To prepare clustered cTB blood sample models, single and clustered JEG-3 cells (pre-labeled with DiD and enumerated under an optical microscope) were spiked into the PBMC (pre-labeled with DiD) isolated from 2 ml of healthy non-pregnant female donor's whole blood'. How were single and clustered JEG-3 cells picked and then spike these cells into PBMCs? The details should be mentioned in the methods section.

4) It was mentioned, 'The predominant distribution in the first two channels (81–90%) suggests that NanoVelcro Chips have sufficient channel length to capture both single and clustered JEG-3 cells.', but in the inset of Figure 2c, channels 1, 2 and 3 were not connected. It's unclear whether these 3 channels were connected or not. The chip design should be shown.

5) Related to Figure 4b, the mean value of counts of single, clustered and total cTBs per 2 ml of blood for 274 pregnant women with PAS, placenta previa, and normal placentation should be shown.

Reviewer #2 (Remarks to the Author):

This is an original study assessing the correlation between circulating trophoblast cell clusters (cTBs) in maternal blood and placenta accrete spectrum (PAS). The authors use a novel assay that captures cTBs with a nanostructure-embedded microchip. The chip enriches both single and clustered cTBs from PBMCs purified from peripheral blood of pregnant women. They found clustered cTBs to be particularly associated with PAS. PAS is an important cause of maternal morbidity and mortality and the topic is of interest to readers of the journal. The approach is quite novel and may lead to increased understanding of the pathophysiology of PAS. The assay also may prove useful for prediction of PAS. The paper is well-written and the authors do a nice job of confirming that the cell clusters are truly trophoblast. Questions and suggestions for enhancing this manuscript include:

1. More details regarding the clinical cases are of interest. Did all PAS cases have a hysterectomy? Were all confirmed histologically? Many did not have a prior cesarean, making them unusual for PAS as typically defined. It is not clear how many had both a prior cesarean and a previa.
2. Stratification of cTBs by accreta, increta and percreta would be of interest. Increased signal with worsening disease would strengthen the authors hypothesis.
3. There are no sample size calculations included. It is unclear as to how the number of cases was chosen.
4. A flow diagram of potential subjects and actually enrolled subjects should be included. It is likely that some women declined participation. Were consecutive patients approached or consented? Was there possible bias? Also, many cases were enrolled in early pregnancy. It is likely that some thought to have previa or PAS early in pregnancy did not actually have the condition at delivery.
5. It would be useful to report positive and negative predictive values since the authors propose using this as a screening test. Although these are influenced by the prevalence, they are important markers of the utility of a screening test.
6. It also would be good to show a diagram with positive results in each group. There is clearly overlap in results among groups and visualization of this can help readers assess the utility of the test.
7. The authors should use some of the cohort to develop their prediction model and part of the cohort to provide internal validation. The assay also requires external validation.
8. Given these concerns, the authors overstate their conclusions regarding this assay to predict PAS.
9. The assay likely performs differently in varying gestational age epochs. Tight gestational age windows would allow for more reproducible results.

10. It is important to note that although the ROCs are good, similar ROCs are noted for prediction of PAS with clinical factors alone, ultrasound alone, and those two markers combined. Does the current assay improve prediction if added to existing methods to predict PAS?

11. The assay requires purification of PBMCs, rapid assay and expensive technology. Accordingly, it may be difficult to create a rapid, cost-effective assay that is widely available. The authors should comment on potential scale up and cost of the assay.

12. Was there an a priori plan to exclude cases due to diabetes and / or fetal abnormalities?

13. It is surprising that the assay did not correlate with prior CD. This deserves comment.

Detailed Responses to the Reviewers' Comments

Response to Reviewer #1

Comments: The authors demonstrate that the counts of single, clustered and total cTBs in PAS are significantly higher than those in non-PAS groups, and the counts of cTBs can be used to distinguish PAS from normal placentation and/or placenta previa. The paper is generally clear and well prepared, but the authors should address the following comments to further clarify their work:

Response: We thank the reviewer for the positive comments regarding our study. The valuable suggestions and comments are much appreciated and are addressed point-by-point below.

Comment 1: Although the EVT origin of cTBs captured on NanoVelcro Chips was verified by well-validated immunocytochemistry (ICC) markers and confirmed by the detection of trophoblast-specific genes by RT-ddPCR, but they didn't confirm the origin of cTBs from fetuses. This point should be discussed.

Response 1: We appreciate the reviewer for recognizing that our work has successfully validated cTBs' EVT origin. In fact, we (reference 25: ACS Nano 2017, 11, 8167-8177) and other groups (reference 48: *Reprod. Biomed. Online* 2012, 25, 508-520) have previously demonstrated that the cTBs are fetal origin as they carry both paternal and maternal alleles.

Following the reviewer's suggestion, we incorporated the description and references in the discussion section on **page 26, lines 479–482** of the revised manuscript, which reads as follows:

"In previous studies^{25,48}, cTBs have been demonstrated to be of fetus origin as they carry both paternal and maternal alleles. In this study, we further confirmed they are of trophoblast origin, similar to recent studies that identified cTBs to be EVTs of placental origin^{20,21}."

Comment 2: The clone number of antibodies including anti-HLA-G, anti-CK7 and the companies from which they were purchased should be added.

Response 2: Following the reviewer's suggestion, the clone number of antibodies including anti-HLA-G, anti-CK7 and the companies from which they were purchased were incorporated into the methods section of revised manuscript (**page 31, lines 607–610**), which reads as follows:

“DAPI (#422801, BioLegend, USA), anti-cytokeratin 7 antibody (Rabbit polyclonal IgG) (#ab53123, abcam, USA), anti-HLA-G antibody (4H84) (#ab52455, abcam, USA), and anti-CD45 monoclonal antibody (YAML501.4) (#MA5-17687, Thermofisher, USA) for identification of cTBs (DAPI+/CK7+/HLA-G+/CD45-) and cTB-clusters.”

Comment 3: In the methods section, it was mentioned, ‘To prepare clustered cTB blood sample models, single and clustered JEG-3 cells (pre-labeled with DiD and enumerated under an optical microscope) were spiked into the PBMC (pre-labeled with DiD) isolated from 2 ml of healthy non-pregnant female donor’s whole blood’. How were single and clustered JEG-3 cells picked and then spike these cells into PBMCs? The details should be mentioned in the methods section.

Response 3: We thank the reviewer for this comment and suggestion. Please allow us to clarify that the single and clustered JEG-3 cells were not picked separately. By culturing JEG-3 cells under sphere-forming conditions (reference 34: Cell. Physiol. Biochem. 2016, 39, 1421-1432), a floating mixture of single and clustered JEG-3 cells was obtained. After dye labeling, the cell mixture was resuspended with PBS. We adopted a similar protocol that was reported by A. F. Sarioglu *et al.* (reference 50: Nat. Methods 2015, 12, 685-691) to count single and clustered JEG-3 cells in the mixture by acquiring fluorescence microscope images before spiking and then spike the mixture of single and clustered JEG-3 cells into PBMCs to prepare a clustered cTB blood sample model for the immobilization onto a NanoVelcro Chip. The details have been added to the methods section (**pages 29–30, lines 545–573**) of the revised manuscript, which now reads as follows:

“A floating mixture of single and clustered JEG-3 cells were generated under a sphere-forming condition³⁴...Then, the mixture of single and clustered JEG-3 cells was pre-stained with Vybrant™ DiO green fluorescent dye, and the PBMCs isolated from healthy non-pregnant female donor’s whole blood were pre-stained with Vybrant™ DiD red fluorescent dye (Invitrogen, USA) ... After dye labeling, the cell mixture was resuspended with PBS. A 2.5-μL mixture of single and clustered JEG-3 cells was deposited on an ultralow-attachment culture dish⁵⁰. The population of single and clustered JEG-3 cells was characterized by acquiring fluorescence microscope images before spiking. After that, the mixture of single and clustered JEG-3 cells was re-pipetted and spiked into PBMCs to prepare a clustered cTB blood sample model for the immobilization onto a NanoVelcro Chip. The culture dish was reimaged to account for the mixture of single and clustered JEG-3 cells that remained attached to the

surface. By post-processing these microscope images, single JEG-3 cells and clustered JEG-3 cells within each cluster that were ultimately spiked into female donor's whole blood could be accurately enumerated."

Comment 4: It was mentioned, 'The predominant distribution in the first two channels (81–90%) suggests that NanoVelcro Chips have sufficient channel length to capture both single and clustered JEG-3 cells.', but in the inset of Figure 2c, channels 1, 2 and 3 were not connected. It's unclear whether these 3 channels were connected or not. The chip design should be shown.

Response 4: We thank the reviewer for pointing out this figure issue. These three channels were connected. We revised the chip design in the **inset of Figure 2c** (also see below) in the revised manuscript. Besides, we provided with a photograph and schematic diagram showing the entire NanoVelcro device in **Supplementary Figure 1** (also see below) in the revised supplementary information.

Figure 2c. Inset: a photograph of NanoVelcro Chip showing the connected three channels.

Supplementary Figure 1. A photograph and schematic diagram showing the entire NanoVelcro device.

Comment 5: Related to Figure 4b, the mean value of counts of single, clustered and total cTBs per 2 ml of blood for the pregnant women with PAS, placenta previa, and normal placentation should be shown.

Response 5: We thank the reviewer for this suggestion. We have incorporated the mean value in the figure legend (page 17, lines 295–298) of Figure 4b which is now **Figure 5b–d** in the revised manuscript. Because of the highly skewed distribution of cTB counts for PAS, placenta previa, and normal placentation, we used a log₂-transformation in order to make the data follow the normal distribution for one-way ANOVA analysis. Now the log₂-transformed mean for PAS, placenta previa, and normal placentation are shown in the **Figure 5b–d** (also see below) in the revised manuscript. Since we are using the single cTBs, clustered cTBs and cTB-clusters in the logistic regression model for predicting PAS in the revised manuscript, the total cTBs (i.e., single cTBs plus clustered cTBs) are replaced by the cTB-clusters in the revised **Figure 5d**.

Figure 5. Comparison of single and clustered cTBs, as well as cTB-clusters in PAS, placenta previa, and normal placentation... Counts of: **b** single cTBs, **c** clustered cTBs, and **d** cTB-clusters per 2 ml of blood for pregnant women with PAS ($n = 65$), placenta previa ($n = 59$), and normal placentation ($n = 44$) are shown. Counts were log₂-transformed. The two shorter horizontal lines denote the 25–75% interquartile ranges (IQR) and the longer horizontal lines in between denote the median. Data are expressed as Mean \pm SE for **b** single cTBs: PAS (4.4 ± 0.2), placenta previa (2.2 ± 0.2), normal placentation (1.6 ± 0.2); **c** clustered cTBs: PAS (4.7 ± 0.3), placenta previa (0.6 ± 0.2), normal placentation (0.3 ± 0.1); and **d** cTB-clusters: PAS (3.0 ± 0.2), placenta previa (0.3 ± 0.1), normal placentation (0.2 ± 0.1). Source data are provided in the Source Data file.

Response to Reviewer #2

Comments: This is an original study assessing the correlation between circulating trophoblast cell clusters (cTBs) in maternal blood and placenta accreta spectrum (PAS). The authors use a novel assay that captures cTBs with a nanostructure-embedded microchip. The chip enriches both single and clustered cTBs from PBMCs purified from peripheral blood of pregnant women. They found clustered cTBs to be particularly associated with PAS. PAS is an important cause of maternal morbidity and mortality and the topic is of interest to readers of the journal. The approach is quite novel and may lead to increased understanding of the pathophysiology of PAS. The assay also may prove useful for prediction of PAS. The paper is well-written and the authors do a nice job of confirming that the cell clusters are truly trophoblast. Questions and suggestions for enhancing this manuscript include:

Response: We appreciate the reviewer for his/her thoughtful comments and suggestions. Please see our point-by-point responses below.

Comment 1: More details regarding the clinical cases are of interest. Did all PAS cases have a hysterectomy? Were all confirmed histologically? Many did not have a prior cesarean, making them unusual for PAS as typically defined. It is not clear how many had both a prior cesarean and a previa.

Response 1: Following the reviewer's suggestion, we incorporated more details regarding the clinical cases and summarized the clinical information in **Table 1** and **Supplementary Table 3** (also see below) in the revised manuscript and supplementary information.

Not all PAS cases have a hysterectomy. The patients who underwent hysterectomy were confirmed histologically. The patients who didn't have hysterectomy were confirmed PAS during the cesarean delivery by FIGO classification for clinical diagnosis. The confirmation of PAS by histology or operative findings are included in the methods section on **page 28, lines 523–528** of the revised manuscript, which now reads as follows:

"PAS was confirmed by histopathological diagnosis after delivery after reviewed by pathologists with expertise in gynecological and perinatal pathology as part of routine clinical care for those patients who had a hysterectomy. The patients who did not have a hysterectomy were confirmed PAS during the cesarean delivery⁷. An intraoperative or clinical diagnosis of PAS is

made in accordance with the International FIGO classification of PAS based on general classification and grading³ (**Supplementary Table 3**).”

More information about prior cesarean and previa have been summarized in the revised manuscript (**page 20, lines 353–355**), which now reads as follows:

“Among all 63 cases of PAS, there are 43 (66%) patients who have both a previous CD and a previa and 5 (8%) patients have neither a previous CD nor a previa (**Supplementary Table 3**), which are the most important two risk factors for PAS.”

Table 1. Clinical information for pregnant women (n = 168) enrolled in the study.

Characteristics	PAS (n = 65)	Non-PAS (n = 103)		p value (PAS versus non-PAS)
		Placenta previa (n = 59)	Normal placentation (n = 44)	
Median maternal age (range)-yo	36 (23–44)	35 (19–54)	37 (22–45)	0.877
Pre-pregnancy BMI* (range)	24 (16–49)	23 (17–32)	24 (19–44)	0.486
In vitro fertilization (IVF)-n (%)				0.583
Yes	14 (21.5)	11 (18.6)	15 (34.1)	
No	51 (78.5)	48 (81.4)	29 (65.9)	
Gravidity-n (%)				0.018
1	8 (12.3)	7 (11.9)	15 (34.1)	
2	11 (16.9)	16 (27.1)	15 (34.1)	
≥3	46 (70.8)	36 (61.0)	14 (31.8)	
Parity-n (%)				0.029
0	10 (15.4)	13 (22.0)	20 (45.5)	
1	26 (40.0)	19 (32.2)	21 (47.7)	
≥2	29 (44.6)	27 (45.8)	3 (6.8)	
Previous cesarean delivery (CD)-n (%)				0.001
0	15 (23.1)	21 (35.6)	32 (72.7)	
1	30 (46.1)	20 (33.9)	11 (25.0)	
≥2	20 (30.8)	18 (30.5)	1 (2.3)	

*BMI = body mass index (kg/m²)

Supplementary Table 3. Summary of PAS patients with/without placenta previa, previous cesarean delivery (CD), and hysterectomy.

Previa	Previous CD	Cases (%)	Hysterectomy	Cases
Yes	Yes	43 (66%)	Yes	32 (49%)
Yes	No	10 (15%)	No	33 (51%)
No	Yes	7 (11%)		
No	No	5 (8%)		
Total PAS			65	

Comment 2: Stratification of cTBs by accreta, increta and percreta would be of interest. Increased signal with worsening disease would strengthen the authors hypothesis.

Response 2: Following the reviewer’s comments, we have tried the stratification of single cTBs, clustered cTBs, and total cTBs by accreta, increta and percreta. There is a trend showing increased signal with worsening disease. However, the comparison is not statistically significant ($p > 0.05$). The data are summarized in **Supplementary Figure 4** (also see below). Results are discussed on **page 16, lines 274–279** of the revised manuscript, which now reads:

“This result raises the question of the correlation between cTB enumeration and severity of the disease. Therefore, we first compared the counts of single and clustered cTBs, as well as cTB-clusters among the subtypes of PAS (i.e., accreta versus increta and percreta). Results summarized in **Supplementary Fig. 4** indicated that despite a trend showing increased numbers of cTBs or cTB-clusters with more severe disease, the comparison is not statistically significant ($p > 0.05$) in the current study.”

Supplementary Figure 4. Counts of single and clustered cTBs as well as cTB-clusters based on stratification of placenta accreta spectrum (PAS) by accreta, increta and percreta: **a** single cTBs, **b** clustered cTBs, **c** cTB-clusters. Counts were log2-transformed. The two shorter horizontal lines denote the 25–75% interquartile ranges (IQR) and the longer horizontal lines

in between denote the median Data are expressed as Mean \pm SE for a single cTBs: accreta (4.2 \pm 0.2), increta and pecreta (4.6 \pm 0.3); b clustered cTBs: accreta (4.5 \pm 0.4), increta and pecreta (4.9 \pm 0.5); and c cTB-clusters: accreta (2.9 \pm 0.3), increta and pecreta (3.1 \pm 0.3). Source data are provided in the Source Data file.

Comment 3: There are no sample size calculations included. It is unclear as to how the number of cases was chosen.

Response 3: Following the reviewer's comments, we incorporated the sample size calculation on **page 28, lines 534–540** of the revised manuscript, which reads as follows:

“The sample size was calculated according to the AUC comparison between our assay and the clinical ultrasound using the paired DeLong's test. A sample size of 128 (51 cases of PAS and 77 cases of control, the ratio of sample sizes in negative/positive groups is 3/2) is expected to have 80% power to detect differences between the AUCs for cTB assay versus ultrasound, assuming AUC = 0.920 for our cTB assay, AUC = 0.800 for ultrasound, when a correlation between the assays of 0.5 was assumed. The power was obtained for a two-sided test at 0.05 significance level.”

Comment 4: A flow diagram of potential subjects and actually enrolled subjects should be included. It is likely that some women declined participation. Were consecutive patients approached or consented? Was there possible bias? Also, many cases were enrolled in early pregnancy. It is likely that some thought to have previa or PAS early in pregnancy did not actually have the condition at delivery.

Response 4: Following the reviewer's suggestion, we included a flow diagram of potential subjects and actually enrolled subjects in **Figure 3** (also see below) in the revised manuscript. There is no declination of participation in this study, and consecutive patients were consented in this study. We agree with the reviewer's comment that many cases were enrolled in early pregnancy. It is likely that some thought to have previa or PAS early in pregnancy did not actually have the condition at delivery. We allocated subjects to the final PAS group or non-PAS group according to the final pathology or clinical diagnosis at delivery in order to remove the bias of mis-diagnosed PAS prior to delivery.

The related contents can be found on **page 11, lines 223–225** of the revised manuscript, which reads as follows:

“Fig. 3 depicts our clinical study design. Among the 171 eligible pregnant women recruited in this study, three subjects were excluded due to fetal genetic/congenital anomalies, or technical failure.”

Figure 3. Clinical study design flowchart depicting the recruitment and exclusions from the study cohort. Blood samples from 109 and 62 eligible study subjects gestational age (GA, 6–40 weeks) were collected and processed in Los Angeles, USA, and Shenzhen, China, respectively. After 3 samples were excluded due to fetal genetic/congenital anomalies or technical failure, blood samples from a final cohort of 168 pregnant women were subjected to NanoVelcro Chip assays and analyzed according to the combination of single and clustered cTBs, as well as cTB-clusters.

Comment 5: It would be useful to report positive and negative predictive values since the authors propose using this as a screening test. Although these are influenced by the prevalence, they are important markers of the utility of a screening test.

Response 5: We thank the reviewer for this clinically relevant suggestion. Following the reviewer’s suggestion, we have added the positive and negative predictive values as well as the data entered in a 2x2 table for each analysis in **Supplementary Figure 6** (also see below).

Results are also summarized **page 18, lines 315–319** of the revised manuscript, which reads as follows:

*“We observed that the PAS prevalence is higher in Shenzhen cohort (47.5%) than the USA cohort (33.6%) in this study. We next calculated the positive prediction value (PPV) and negative prediction value (NPV) for the cTB assay in the subpopulations of USA cohort (PPV = 85.3%, NPV = 90.4%) and Shenzhen cohort (PPV = 92.9%, NPV = 90.9%) as well as all cohorts (PPV = 83.8%, NPV = 92.0%). These data are summarized in **Supplementary Fig. 6.**”*

Supplementary Figure 6. The positive predictive values (PPV) and negative predictive values (NPV) as well as sensitivity and specificity for **a** USA cohort, **b** Shenzhen cohort and **c** all cohorts. The data was entered in a 2x2 table for each analysis.

Comment 6: It also would be good to show a diagram with positive results in each group. There is clearly overlap in results among groups and visualization of this can help readers assess the utility of the test.

Response 6: Following the reviewer’s suggestion, we have showed diagrams with positive results and negative results in a 2x2 table for the cTB assay in **Supplementary Figure 6** in the revised supplementary information. Overall, there are 57 positive results in PAS group and 11 positive results in the Non-PAS group.

Comment 7: The authors should use some of the cohort to develop their prediction model and part of the cohort to provide internal validation. The assay also requires external validation.

Response 7: Following the reviewer’s suggestion, we tried our best to recruit more patients in the past 7 months. As of data cutoff, we have recruited 168 patients overall. We have developed the prediction model and conducted the internal validation (leave-one-out cross validation) and

external validation (independent test set). The study design is detailed in the flowchart in **Figure 3**. The receiver operating characteristic (ROC) curves and AUCs for the training set, cross validation set, and external validation test set are summarized in **Figure 6a–c** (also see below). Results are also included on **page 18, lines 306–314** in the revised manuscript, which reads as follows:

“We then conducted a stepwise multivariate logistic regression model (**Supplementary Table 2**) to combine the single and clustered cTBs, as well as cTB-clusters for differentiating PAS from non-PAS (placenta previa plus normal placentation) followed by the leave-one-out cross-validation, and applied this logistic regression model to an independent test set (**Fig. 3**). **Fig. 6a–c** summarizes the diagnostic performance of the combined cTB assay for distinguishing PAS from non-PAS in the training set, after leave-one-out cross validation, and in the independent test set with areas under the curve (AUC) of 0.947 (sensitivity = 88.9%, specificity = 87.5%), 0.946 (sensitivity = 88.9%, specificity = 87.5%), and 0.926 (sensitivity = 90.0%, specificity = 90.3%), respectively.”

Figure 6. ROC curves of cTB assay analyzed in **a** the training set, **b** after leave-one-out cross validation, and **c** Independent test set for distinguishing PAS from non-PAS during all GA (6–40 weeks). Area under the curves (AUC) with the sensitivity and specificity of the assays at the optimal cutoffs are listed for each graph.

Comment 8: Given these concerns, the authors overstate their conclusions regarding this assay to predict PAS.

Response 8: Following the reviewer’s comment, we have accumulated more patient data and try our best to address these concerns. We have also revised our conclusion to avoid

overstatement on **page 27, lines 503–506** of the revised manuscript, which now reads as follows:

“We envision that the combination of cTBs and cTB-clusters captured on the NanoVelcro Chips for detecting PAS early in gestation will enable a promising quantitative assay to serve as a noninvasive test and also as a complement to ultrasonography to improve diagnostic accuracy for PAS early in gestation.”

Comment 9: The assay likely performs differently in varying gestational age epochs. Tight gestational age windows would allow for more reproducible results.

Response 9: We agree with the the reviewer’s comments that tight gestational age windows would allow for more reproducible results. We analyzed the subgroups of the patients with earlier GA (GA < 24 weeks) and those with late GA (GA ≥ 24 weeks), respectively. The results of subgroups with tight GA windows (**Figure 6e, f**, also see below) showed reproducible diagnostic performance with similar AUCs compared to the AUCs for all GA (**Figure 6d**).

Figure 6. Comparison of ROC curves of cTB assay with or without ultrasound for distinguishing PAS from non-PAS in **d** all GA (6–40 weeks), **e** earlier GA (The first and second trimester, < 24 weeks), and **f** late GA (≥ 24 weeks). AUC for each diagnostic model was listed for each ROC curve. Source data are provided in the Source Data file.

Results are also included on **page 19, lines 336–349** of the revised manuscript, which reads as follows:

*“To explore whether the cTB assay performs differently in varying GA, we then conducted logistic regression models (**Supplementary Table 2**) and comparison of ROC analysis on the subpopulations of pregnant women at earlier GA (< 24 weeks) and late GA (≥ 24 weeks) to distinguish PAS from non-PAS, respectively. The cTB assay in the subpopulations can*

differentiate PAS from non-PAS regardless of GA. Again, the cTB assay can improve prediction of PAS if combined with the ultrasound in both earlier GA and late GA. In the earlier GA population (Fig. 6e), the combination of cTB assay along with ultrasound achieved an AUC of 0.976, which outperformed the cTB assay alone (with the AUC of 0.976 versus 0.924, $p = 0.171$, not significant), or the ultrasound alone (with the AUC of 0.976 versus 0.826, $p = 0.002$). In the late GA population (Fig. 6f), the combination of cTB assay along with ultrasound achieved an AUC of 0.979, which outperformed the cTB assay alone (with the AUC of 0.979 versus 0.961, $p = 0.255$, not significant), or the ultrasound alone (with the AUC of 0.979 versus 0.884, $p = 0.001$). These results show that the potential of cTB assay for the detection of PAS can be used throughout gestation, and the cTB assay is also reproducible in tight GA windows of both earlier GA and late GA.”

Comment 10: It is important to note that although the ROCs are good, similar ROCs are noted for prediction of PAS with clinical factors alone, ultrasound alone, and those two markers combined. Does the current assay improve prediction if added to existing methods to predict PAS?

Response 10: Following the reviewer’s comments, we have conducted the stepwise multivariate logistic regression analysis to see which clinical factor and/or which predictor combination would be the best for the prediction of PAS. The results summarized in **Supplementary Table S2** (also see below) and **Figure 6d** showed that current cTB assay improved the prediction of PAS if combined with ultrasound. Results are also included on **pages 18–19, lines 320–330** of the revised manuscript, which reads as follows:

*“To explore whether the cTB assay improves prediction of PAS if combined with ultrasound and other clinical risk factors listed in **Table 1**, such as a previous cesarean delivery (CD), maternal age, maternal body mass index (BMI), in vitro fertilization (IVF), gravidity and parity, multivariate logistic regression analysis was used to evaluate and select significant clinical predictors. The results showed that in addition to the cTB assay, ultrasound is the most statistically significant contributor associated with PAS ($p < 0.001$, **Supplementary Table 2**). We then constructed a stepwise multivariate logistic regression model to see if the combination of the cTB assay with ultrasound improves prediction of PAS. As the comparison of ROC curves showed in **Fig. 6d**, the combination of cTB assay along with ultrasound achieved an AUC of 0.978, which outperformed the cTB assay alone (with the AUC of 0.978 versus 0.942, $p = 0.052$, not significant), or the ultrasound alone (with the AUC of 0.978 versus 0.866, $p < 0.001$).”*

Supplementary Table 2. Detailed information for the multivariate logistic regression analysis.

Multivariate logistic regression results (PAS versus non-PAS)			
Logistic regression model-stepwise	Odds ratio	95% Confidence intervals (CI)	p value
cTB assay-training cohort (Fig. 6a, AUC=0.947)			
cTB-clusters	1.710	1.264–2.314	<0.001
Single cTBs	1.130	1.013–1.260	0.028
Clustered cTBs	Not included in the model		
cTB assay-all gestational age (Fig. 6d-blue line, AUC=0.942)			
cTB-clusters	1.760	1.356–2.284	<0.001
Single cTBs	1.093	1.014–1.177	0.020
Clustered cTBs	Not included in the model		
cTB assay+ultrasound-all gestational age (Fig. 6d-orange line, AUC=0.978)			
Clustered cTBs	1.231	1.114–1.360	<0.001
Ultrasound	88.8	18.71–421.4	<0.001
Single cTBs, cTB-clusters	Not included in the model		
cTB assay-earlier gestational age (Fig. 6e-blue line, AUC=0.924)			
cTB-clusters	1.937	1.400–2.679	<0.001
Single cTBs, clustered cTBs	Not included in the model		
cTB assay+ultrasound-earlier gestational age (Fig. 6e-orange line, AUC=0.976)			
cTB-clusters	1.888	1.234–2.887	0.003
Ultrasound	39.76	3.924–402.8	0.002
Single cTBs, clustered cTBs	Not included in the model		
cTB assay-late gestational age (Fig. 6f-blue line, AUC=0.961)			
cTB-clusters	2.090	1.311–3.331	0.002
Single cTBs	1.282	1.078–1.525	0.005
Clustered cTBs	Not included in the model		
cTB assay+ultrasound-late gestational age (Fig. 6f-orange line, AUC=0.979)			
cTB-clusters	3.085	1.493–6.377	0.002
Ultrasound	145.7	12.08–1758	<0.001
Single cTBs, clustered cTBs	Not included in the model		
cTB assay+clinical factors-all gestational age (AUC=0.978)			
Clustered cTBs	1.231	1.114–1.360	<0.001
Ultrasound	88.8	18.71–421.4	<0.001
Single cTBs, cTB-clustereds, Maternal age, BMI, previous CD, Gravidity, Parity, IVF	Not included in the model		

For the stepwise logistic regression model, variables are entered if $p < 0.05$, variables are removed if $p > 0.1$. CD, Cesarean delivery; BMI, body mass index; IVF, in vitro fertilization.

Comment 11: The assay requires purification of PBMCs, rapid assay and expensive technology. Accordingly, it may be difficult to create a rapid, cost-effective assay that is widely available. The authors should comment on potential scale up and cost of the assay.

Response 11: For PBMC purification, we adopted a commercial kit, i.e., SepMate™ PBMC Isolation Tubes from STEMCELL Technologies. The novel design of SepMate™ Tubes enables consistent and hassle-free PBMC isolation in just 15 min. In terms of cTB enumeration by NanoVelcro Chips, the entire workflow is composed of the following steps: (i) capturing cTB cells/clusters onto NanoVelcro Chips (30 min), (ii) overnight immunostaining, and (iii) imaging/counting by a fluorescent microscope (30 min). In our current laboratory setting, 12 samples can be processed by a lab technician in one day. The materials and production cost of a NanoVelcro Chip is < \$10 at UCLA Nanofabrication facility. Once the production of NanoVelcro Chips is scaled up in a commercial setting, the cost is expected to be lower.

Following the reviewer's comment, we supplemented the streamlined workflow in the methods section on **page 31, lines 598–604** and discussed the potential scale up and cost of the assay in the discussion section on **page 25–26, lines 470–472** of the revised manuscript, which reads as follows:

“The PBMCs were isolated in 15 min by using SepMate™ PBMC Isolation Tubes (STEMCELL Technologies, USA) and aliquoted following our previously published protocol⁵¹. PBMCs from 2 mL of whole blood were run through the NanoVelcro Chips using the optimized protocol. The streamlined workflow is composed of the following steps: (i) capturing cTB cells/clusters onto NanoVelcro Chips (30 min), (ii) overnight immunostaining, and (iii) imaging/counting by a fluorescent microscope (30 min).”

“Given the streamlined workflow developed for capturing cTBs and cTB-clusters in NanoVelcro Chips, we envision scaling up this cost-effective assay in a large-scale multicenter validation study.”

Comment 12: Was there an a priori plan to exclude cases due to diabetes and/or fetal abnormalities?

Response 12: Yes, we have a detailed protocol describing the inclusion and exclusion criteria for this study. Our prior plan was to exclude fetal genetic and congenital anomalies as the placenta from these pregnancies would likely be genetically abnormal, affecting the characteristics of circulating trophoblasts from these patients. Moreover, it was reported that the numbers of maternal circulating trophoblasts increased in patients with fetal genetic and congenital anomalies (reference 25: ACS Nano 2017, 11, 8167-8177), which will bias and complicate the enumeration analysis in PAS study. We don't exclude diabetes in this study according to our prior plan. We provided our original IRB protocols and prior study plan as supplementary materials per the editorial office's requirement.

The related contents can be found on **page 11, lines 223–225** of the revised manuscript, which reads as follows:

“Fig. 3 depicts our clinical study design. Among the 171 eligible pregnant women recruited in this study, three subjects were excluded due to fetal genetic/congenital anomalies, or technical failure.”

Comment 13: It is surprising that the assay did not correlate with prior CD. This deserves comment.

Response 13: Following the reviewer's comments, we have summarized the updated statistical results of the Spearman's Rank-Order Correlation analysis to correlate the assay with prior CD in the revised manuscript (**page 20, lines 350–351**), which reads as follows:

“In this study, the numbers of single ($p = 0.010$, $r_s = 0.197$) and clustered cTBs ($p = 0.015$, $r_s = 0.187$), as well as cTB-clusters ($p = 0.015$, $r_s = 0.187$) are correlated with previous CD.”

REVIEWERS' COMMENTS

Reviewer #1 (Remarks to the Author):

This resubmission has clarified my questions from the previous review. I have no further questions and believe the manuscript warrants publication as is.

Reviewer #2 (Remarks to the Author):

The authors have done an admirable job of responding to all of the many comments and questions in an effort to improve their manuscript.

Detailed Responses to the Reviewers' Comments

Response to Reviewer #1

Comments: This resubmission has clarified my questions from the previous review. I have no further questions and believe the manuscript warrants publication as is.

Response: We thank the reviewer very much for his/her positive comments and kind recommendation.

Response to Reviewer #2

Comments: The authors have done an admirable job of responding to all of the many comments and questions in an effort to improve their manuscript.

Response: We appreciate the reviewer for his/her high evaluation of our study.